# LAYER-ADAPTIVE SPARSITY FOR THE MAGNITUDE-BASED PRUNING

**Jaeho Lee[E]**   **Sejun Park[A]**   **Sangwoo Mo[E]**   **Sungsoo Ahn[M]** *   **Jinwoo Shin[Æ]**
[E]KAIST EE   [A]KAIST AI   [M]MBZUAI
{jaeho-lee,sejun.park,swmo,jinwoos}@kaist.ac.kr,peter.ahn@mbzuai.ac.ae

## ABSTRACT

Recent discoveries on neural network pruning reveal that, with a carefully chosen *layerwise sparsity*, a simple magnitude-based pruning achieves state-of-the-art tradeoff between sparsity and performance. However, without a clear consensus on "how to choose," the layerwise sparsities are mostly selected algorithm-by-algorithm, often resorting to handcrafted heuristics or an extensive hyperparameter search. To fill this gap, we propose a novel importance score for global pruning, coined layer-adaptive magnitude-based pruning (LAMP) score; the score is a rescaled version of weight magnitude that incorporates the model-level $\ell_2$ distortion incurred by pruning, and does not require any hyperparameter tuning or heavy computation. Under various image classification setups, LAMP consistently outperforms popular existing schemes for layerwise sparsity selection. Furthermore, we observe that LAMP continues to outperform baselines even in weight-rewinding setups, while the connectivity-oriented layerwise sparsity (the strongest baseline overall) performs worse than a simple global magnitude-based pruning in this case. Code: https://github.com/jaeho-lee/layer-adaptive-sparsity

## 1   INTRODUCTION

Neural network pruning is an art of removing "unimportant weights" from a model, with an intention to meet practical constraints (Han et al., 2015), mitigate overfitting (Hanson & Pratt, 1988), enhance interpretability (Mozer & Smolensky, 1988), or deepen our understanding on neural network training (Frankle & Carbin, 2019). Yet, the *importance of weight* is still a vaguely defined notion, and thus a wide range of pruning algorithms based on various importance scores has been proposed. One popular approach is to estimate the loss increment from removing the target weight to use as an importance score, e.g., Hessian-based approximations (LeCun et al., 1989; Hassibi & Stork, 1993; Dong et al., 2017), coreset-based estimates (Baykal et al., 2019; Mussay et al., 2020), convex optimization (Aghasi et al., 2017), and operator distortion (Park et al., 2020). Other approaches include on-the-fly[1] regularization (Louizos et al., 2018; Xiao et al., 2019), Bayesian methods (Molchanov et al., 2017; Louizos et al., 2017; Dai et al., 2018), and reinforcement learning (Lin et al., 2017).

Recent discoveries (Gale et al., 2019; Evci et al., 2020) demonstrate that, given an appropriate choice of *layerwise sparsity*, simply pruning on the basis of weight magnitude yields a surprisingly powerful unstructured pruning scheme. For instance, Gale et al. (2019) evaluates the performance of magnitude-based pruning (MP; Han et al. (2015); Zhu & Gupta (2018)) with an extensive hyperparameter tuning, and shows that MP achieves comparable or better performance than state-of-the-art pruning algorithms that use more complicated importance scores. To arrive at such a performance level, the authors introduce the following handcrafted heuristic: Leave the first convolutional layer fully dense, and prune up to only 80% of weights from the last fully-connected layer; the heuristic is motivated by the sparsity pattern from other state-of-the-art algorithms (Molchanov et al., 2017) and additional experimental/architectural observations.

Unfortunately, there is an apparent lack of consensus on "how to choose the layerwise sparsity" for the magnitude-based pruning. Instead, the layerwise sparsity is selected mostly on an algorithm-by-algorithm basis. One common method is the *global MP* criteria (see, e.g., Morcos et al. (2019)),

---

*Work done at KAIST

[1]i.e., simultaneously training and pruning

$$
\begin{bmatrix} u_1 \\ u_2 \\ u_3 \\ \cdots \end{bmatrix}
\begin{bmatrix} v_1 \\ v_2 \\ v_3 \\ \cdots \end{bmatrix}
\longrightarrow
\begin{bmatrix} \frac{u_1^2}{u_1^2} \\ \frac{u_2^2}{u_1^2 + u_2^2} \\ \frac{u_3^2}{u_1^2 + u_2^2 + u_3^2} \\ \cdots \end{bmatrix}
\begin{bmatrix} \frac{v_1^2}{v_1^2} \\ \frac{v_2^2}{v_1^2 + v_2^2} \\ \frac{v_3^2}{v_1^2 + v_2^2 + v_3^2} \\ \cdots \end{bmatrix}
\longrightarrow
\begin{bmatrix} \frac{u_1^2}{u_1^2} & \frac{v_1^2}{v_1^2} \\ \frac{u_2^2}{u_1^2 + u_2^2} & \varnothing \\ \varnothing & \varnothing \\ \cdots & \cdots \end{bmatrix}
$$

(sorted) Weight Magnitudes      LAMP score      Global Pooling and Pruning

Figure 1: The LAMP score is a squared weight magnitude, normalized by the sum of all "surviving weights" in the layer. Global pruning by LAMP is equivalent to the layerwise magnitude-based pruning with an automatically chosen layerwise sparsity.

where the layerwise sparsity is automatically determined by using a single global threshold on weight magnitude. Lin et al. (2020) propose a magnitude-based pruning algorithm using a feedback signal, using a heuristic rule of keeping the last fully connected layer dense. A recent work by Evci et al. (2020) proposes a magnitude-based dynamic sparse training method, adopting layerwise sparsity inspired from the network science approach toward neural network pruning (Mocanu et al., 2018).

**Contributions.** In search of a "go-to" layerwise sparsity for MP, we take a *model-level distortion minimization* perspective towards MP. We build on the observation of Dong et al. (2017); Park et al. (2020) that each neural network layer can be viewed as an operator, and MP is a choice that incurs minimum $\ell_2$ distortion to the operator output (given a worst-case input signal). We bring the perspective further to examine the "model-level" distortion incurred by pruning a layer; preceding layers scale the input signal to the target layer, and succeeding layers scale the output distortion.

Based on the distortion minimization framework, we propose a novel importance score for global pruning, coined LAMP (Layer-Adaptive Magnitude-based Pruning). The LAMP score is a rescaled weight magnitude, approximating the model-level distortion from pruning. Importantly, the LAMP score is designed to approximate the distortion on the *model being pruned*, i.e., all connections with a smaller LAMP score than the target weight is already pruned. Global pruning[2] with the LAMP score is equivalent to the MP with an automatically determined layerwise sparsity. At the same time, pruning with LAMP keeps the benefits of MP intact; the LAMP score is efficiently computable, hyperparameter-free, and does not rely on any model-specific knowledge.

We validate the effectiveness of LAMP under a diverse experimental setup, encompassing various convolutional neural network architectures (VGG-16, ResNet-18/34, DenseNet-121, EfficientNet-B0) and various image datasets (CIFAR-10/100, SVHN, Restricted ImageNet). In all considered setups, LAMP consistently outperforms the baseline layerwise sparsity selection schemes. We also perform additional ablation studies with one-shot pruning and weight-rewinding setup to confirm that LAMP performs reliably well under a wider range of scenarios.

**Organization.** In Section 2, we briefly describe existing methods to choose the layerwise sparsity for magnitude-based pruning. In Section 3, we formally introduce LAMP and describe how the $\ell_2$ distortion minimization perspective motivates the LAMP score. In Section 4, we empirically validate the effectiveness and versatility of LAMP. In Section 5, we take a closer look at the layerwise sparsity discovered by LAMP and compare with baseline methods and previously proposed handcrafted heuristics. In Section 6, we summarize our findings and discuss future directions. Appendices include the experimental details (Appendix A), complexity analysis (Appendix B), derivation of the LAMP score (Appendix C), additional experiments on Transformer (Appendix D), and detailed experimental results with standard deviations (Appendix E).

## 2    RELATED WORK

This section gives a (necessarily non-exhaustive) survey of various layerwise sparsity selection schemes used for magnitude-based pruning algorithms. Magnitude-based pruning of neural networks dates back to the early works of Janowsky (1989); LeCun et al. (1989), and has been actively studied

---

[2]i.e., using a global threshold for LAMP score

again under the context of model compression since the work of Han et al. (2015). In Han et al. (2015), the authors propose an iterative pruning scheme where the layerwise pruning threshold is determined by the standard-deviation-based heuristic. Zhu & Gupta (2018) propose a uniform pruning algorithm with a carefully tuned gradual pruning schedule combined with weight re-growths. Gale et al. (2019) refine the algorithm by adding a heuristic constraint of keeping the first convolutional layer fully dense and keeping at least 20% of the weights surviving in the last fully-connected layer.

MP has also been widely used in the context of "pruning at initialization." Frankle & Carbin (2019) combine MP with weight rewinding to discover efficiently trainable subnetworks: for small nets, the authors employ uniform layerwise sparsity, but use different rates for convolutional layers and fully-connected layers (with an added heuristic on the last fully-connected layer); for larger nets, authors use global MP. Morcos et al. (2019) consider transferring the "winning ticket" initializations, using the global MP. Evci et al. (2020) proposes a training scheme for sparsely initialized neural networks, where the layerwise sparsity is given by the Erdős-Rényi kernel method; the method generalizes the scheme initially proposed by Mocanu et al. (2018) to convolutional neural networks.

We note that there is a line of results on the *trainable layerwise sparsity*; we refer the interested readers to the recent work of Kusupati et al. (2020) for a concise survey. However, we do not make direct comparisons to these methods, as our primary purpose is to deliver an easy-to-use layerwise sparsity selection scheme without requiring the modification of training objective, or an extensive hyperparameter tuning.

We also note that we focus on the *unstructured sparsity*. While such unstructured pruning techniques have been considered less practical (compared to structured pruning), several recent breakthroughs provide promising methods to bridge this gap; see Gale et al. (2020); Elsen et al. (2020).

## 3 LAYER-ADAPTIVE MAGNITUDE-BASED PRUNING (LAMP)

We now formally introduce the Layer-Adaptive Magnitude-based Pruning (LAMP) score. Consider a depth-$d$ feedforward neural network with weight tensors $W^{(1)}, \ldots, W^{(d)}$ associated with each fully-connected/convolutional layer. For fully-connected layers, corresponding weight tensors are two-dimensional matrices, and for 2d convolutional layers, corresponding tensors are four-dimensional. To give a unified definition of the LAMP score for both fully-connected and convolutional layers, we assume that each weight tensor is *unrolled* (or flattened) to a one-dimensional vector. For each of these unrolled vectors, we assume (without loss of generality) that the weights are sorted in an ascending order according to the given index map, i.e., $|W[u]| \leq |W[v]|$ holds whenever $u < v$, where $W[u]$ denote the entry of $W$ mapped by the index $u$.[3]

The LAMP score for the $u$-th index of the weight tensor $W$ is then defined as

$$\mathsf{score}(u; W) := \frac{(W[u])^2}{\sum_{v \geq u} (W[v])^2}. \tag{1}$$

Informally, the LAMP score (Eq. 1) measures the relative importance of the target connection among all *surviving connections* belonging to the same layer, where the connections with a smaller weight magnitude (in the same layer) have already been pruned. As a consequence, two connections with identical weight magnitudes have different LAMP scores, depending on the index map being used.

Once the LAMP score is computed, we globally prune the connections with smallest LAMP scores until the desired global sparsity constraint is met; the procedure is equivalent to performing MP with an automatically selected layerwise sparsity. To see this, it suffices to check that

$$(W[u])^2 > (W[v])^2 \Rightarrow \mathsf{score}(u; W) > \mathsf{score}(v; W) \tag{2}$$

holds for any weight tensor $W$ and a pair of indices $u, v$. From the definition of the LAMP score (Eq. 1), it is easy to see that the logical relation (2) holds. Indeed, for the connection with a larger weight magnitude, the denominator of Eq. 1 is smaller, while the numerator is larger.

We note that the global pruning with respect to the LAMP score is not identical to the global pruning with respect to the magnitude score $|W[u]|$ (or $(W[u])^2$, equivalently). Indeed, in each layer, there

---

[3]This "order" among weights is required to handle the weights with equal magnitude.

exists exactly one connection with the LAMP score of 1, which is the maximum LAMP score possible. In other words, LAMP keeps at least one surviving connection in each layer. The same does not hold for the global pruning with respect to the weight magnitude score.

We also note that the LAMP score is easy-to-use. Similar to the vanilla MP, the LAMP score does not have any hyperparameter to be tuned, and is easily implementable via elementary tensor operations. Furthermore, the LAMP score can be computed with only a minimal computational overhead; the sorting of squared weight magnitudes required to compute the denominator in Eq. 1 is already a part of typical vanilla MP algorithms. For more discussions, see Appendix B.

## 3.1 DESIGN MOTIVATION: MINIMIZING OUTPUT $\ell_2$ DISTORTION

The LAMP score (Eq. 1) is motivated by the following observation: The layerwise MP is the solution of the layerwise minimization of *Frobenius distortion* incurred by pruning, which can be viewed as a *relaxation* of the output $\ell_2$ distortion minimization with respect to the worst-case input. This observation leads us to the speculation "Reducing the pruning-incurred $\ell_2$ distortion of the model output with respect to the worst-case output may be beneficial to the performance of the retrained model (and perhaps that is why MP works well in practice)." This speculation is not entirely new; the optimal brain damage (OBD; (LeCun et al., 1989)) is also designed around a similar philosophy of loss minimization, without a complete understanding on how the benefit of loss minimization seems to pertain *after retraining*.

Nevertheless, we use this speculation as a guideline to design LAMP as a natural extension of layerwise MP to a global pruning scheme with an automatically determined layerwise sparsity. To make arguments a bit more formal, consider a depth-$d$ fully-connected[4] neural net, whose output given the input $x$ is

$$f(x; W^{(1:d)}) = W^{(d)} \sigma(W^{(d-1)} \sigma(\cdots W^{(2)} \sigma(W^{(1)} x) \cdots)), \qquad (3)$$

where $\sigma$ denotes the ReLU activation and $W_i$ denotes the weight matrix for the $i$-th layer, and $W^{(1:d)} = (W^{(1)}, \ldots, W^{(d)})$ denotes the set of weight matrices.

**Viewing MP as a relaxed layerwise $\ell_2$ distortion minimization.** We first focus on a single fully-connected layer (instead of a full model), and consider the problem of minimizing the pruning-incurred $\ell_2$ distortion in the layer output, given the worst-case input signal. We then observe that the problem can be relaxed to the minimization of *Frobenius distortion* in the weight tensor, whose solution coincides with the layerwise MP. Formally, let $\xi \in \mathbb{R}^n$ be an input vector to a fully-connected layer with the weight tensor $W \in \mathbb{R}^{m \times n}$. We want to prune the tensor to $\widetilde{W} := M \odot W$, where $M$ is an $m \times n$ binary matrix (i.e., having only 0s and 1s as its entries) satisfying some predefined sparsity constraint $\|M\|_0 \leq \kappa$ imposed by the operational constraints (e.g., model size requirements). We wish to find the *pruning mask* $M$ that incurs the minimum $\ell_2$ distortion in the output given the worst-case $\ell_2$-bounded input, i.e.,

$$\min_{\substack{M \text{ binary} \\ \|M\|_0 \leq \kappa}} \sup_{\|\xi\|_2 \leq 1} \|W\xi - (M \odot W)\xi\|_2. \qquad (4)$$

The minimax distortion (4) upper-bounds the minimum expected $\ell_2$ distortion for any distribution of $\xi$ supported on the unit ball, and thus can be viewed as a data-oblivious version of the pruning algorithms designed for loss minimization (using squared loss). By the definition of the spectral norm,[5] Eq. 4 is equivalent to

$$\min_{\substack{M \text{ binary} \\ \|M\|_0 \leq \kappa}} \|W - M \odot W\|, \qquad (5)$$

where $\|\cdot\|$ denotes the spectral norm. Using the fact that $\|A\| \leq \|A\|_F$ holds for any matrix $A$[6] (where $\|\cdot\|_F$ denotes the Frobenius norm), the optimization (5) can be relaxed to the Frobenius

---

[4]An extension to convolutional neural network is straightforward; see, e.g., (Sedghi et al., 2019).

[5]which is the operator norm with respect to the $\ell_2$ norms, i.e., $\sup_{\xi \neq 0} \frac{\|W\xi\|_2}{\|\xi\|_2}$,

[6]The inequality is a simple consequence of the Cauchy-Schwarz inequality: $\|Ax\|_2^2 = \sum_i (\sum_j A_{ij} x_j)^2 \leq \sum_i (\sum_j A_{ij}^2) \cdot (\sum_j x_j^2) = \|A\|_F^2 \|x\|_2^2$, where subscripts denote weight indices.

distortion minimization

$$\min_{\substack{M \text{ binary} \\ \|M\|_0 \leq \kappa}} \|W - M \odot W\|_F = \min_{\substack{M \text{ binary} \\ \|M\|_0 \leq \kappa}} \sqrt{\sum_{\substack{i \in \{1,\ldots,m\} \\ j \in \{1,\ldots,n\}}} (1 - M_{ij}) W_{ij}^2}, \tag{6}$$

where $W_{ij}, M_{ij}$ denote $(i,j)$-th entries of $W, M$, respectively. From the right-hand side of Eq. 6, we see that the layerwise MP, i.e., setting $M_{ij} = 1$ for $(i,j)$ pairs with top-$\kappa$ largest $W_{ij}$, is the optimal choice to minimize the Frobenius distortion incurred by pruning. This observation motivates us to view the layerwise MP as the (approximate) solution of the output $\ell_2$ distortion minimization procedure, and speculate the connection between the *small output $\ell_2$ distortion* and the *favorable performance of the pruned-retrained subnetwork* (given the unreasonable effectiveness of seemingly-naïve MP as demonstrated by Gale et al. (2019)).

**LAMP: greedy, relaxed minimization of model output distortion.** Building on this speculation, we now ask the following question: "How can we select the layerwise sparsity of MP to have small model-level output distortion?" To formalize, we consider the minimization

$$\min_{\sum_{i=1}^d \|M^{(i)}\|_0 \leq \kappa} \sup_{\|x\|_2 \leq 1} \|f(x; W^{(1:d)}) - f(x; \widetilde{W}^{(1:d)})\|_2, \tag{7}$$

where $\kappa$ denotes the model-level sparsity constraint imposed by the operational requirements and $\widetilde{W}^{(i)} := M^{(i)} \odot W^{(i)}$ denotes the pruned version of the $i$-th layer weight matrix.

Due to the nonlinearities from the activation functions, it is difficult to solve Eq. 7 exactly. Instead, we consider the following *greedy* procedure: At each step, we (a) approximate the distortion incurred by pruning a *single connection*, (b) remove the connection with the smallest score, and then (c) go back to step (a) and re-compute the scores based on the *pruned model*.

Once we assume that only one connection is pruned at a single iteration of the step (a), we can use the following *upper bound* of the model output distortion to relax the optimization (7): With $\widetilde{W}_i$ denoting a pruned version of $W_i$, we have

$$\sup_{\|x\|_2 \leq 1} \|f(x; W^{(1:d)}) - f(x; W^{(1:i-1)}, \widetilde{W}^{(i)}, W^{(i+1:d)})\|_2 \leq \frac{\|W^{(i)} - \widetilde{W}^{(i)}\|_F}{\|W^{(i)}\|_F} \left( \prod_{j=1}^d \|W^{(j)}\|_F \right) \tag{8}$$

(see Appendix C for a derivation). Despite the sub-optimalities from the *relaxation*, considering the right-hand side of Eq. 8 provides two favorable properties. First, the right-hand side is free of any activation function, and is equivalent to the layerwise MP. Second, the score can be computed *in advance*, i.e., does not require re-computing after pruning each connection. In particular, the product term $\prod_{j=1}^d \|W^{(j)}\|_F$ does not affect the pruning decision, and the denominator can be pre-computed with the cumulative sum $\sum_{v \geq u} (W^{(i)}[v])^2$ for each index $u$ for $W^{(i)}$. This computational trick leads us to the LAMP score (1).

## 4 EXPERIMENTS & ANALYSES

To empirically validate the effectiveness of the proposed method, we compare LAMP with following layerwise sparsity selection schemes for magnitude-based pruning:

- **Global.** A global threshold on the weight magnitudes is imposed on every layer to meet the global sparsity constraint, and the layerwise sparsity is automatically determined according to this threshold; see, e.g., Morcos et al. (2019).

- **Uniform.** Every layer is pruned to have identical layerwise sparsity levels, which is in turn equal to the global sparsity constraint; see, e.g., Zhu & Gupta (2018).

- **Uniform+.** Same as Uniform, but we impose two additional constraints: (1) we keep the first convolutional layer unpruned, and (2) retain at least 20% of connections alive in the last fully-connected layer; this heuristic rule is proposed by Gale et al. (2019).

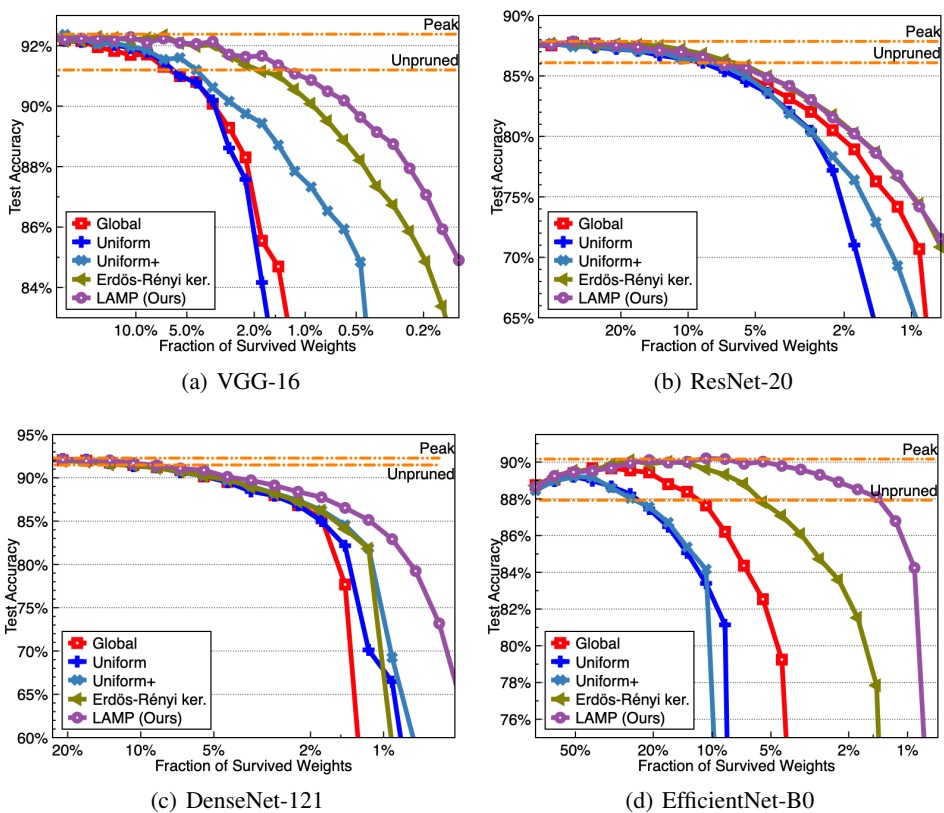

Figure 2: Sparsity-accuracy tradeoff curves of VGG-16, ResNet-18, DenseNet-121, and EfficientNet-B0. All models are iteratively pruned and retrained with CIFAR-10 dataset.

- **Erdős-Rényi kernel.** An extension of Erdős-Rényi method (originally given by Mocanu et al. (2018)) accounting for convolutional layers, as proposed by Evci et al. (2020). The numbers of nonzero parameters of sparse convolutional layers are scaled proportional to $1 - \frac{n^{l-1} + n^l + w^l + h^l}{n^{l-1} \cdot n^l \cdot w^l \cdot h^l}$, where $n^l$ denotes the number of neurons at layer $l$, and $w^l, h^l$ denotes the width and height of the $l$th layer convolutional kernel.

As a default setup, we perform *five independent trials* for each baseline method, where in each trial we use iterative pruning-and-retraining (Han et al., 2015): we prune 20% of the surviving weights at each iteration. For the Restricted-ImageNet experiment, we provide the result from four trials. For a clear presentation, we only report the average on the figures appearing in the main text. Standard deviations for five-seed results in Section 4.1 will be given in Appendix E. Also, in Appendix D, we report additional experimental results for language modeling tasks (Penn Treebank and WT-2) on Transformers (Vaswani et al., 2017).

**Summary of observations.** From the experimental results (Figs. 2 to 4), we observe that LAMP consistently outperforms all other baselines, in terms of the sparsity-accuracy tradeoff. The performance gap between LAMP and baseline methods seems be more pronounced with modern network architectures, e.g., EfficientNet-B0. We also observe that LAMP performs well under weight rewinding setup, while the strongest baseline (Erdős-Rényi kernel) seems to be sensitive to such rewinding.

## 4.1 MAIN RESULTS

Our main experimental results are on image classification models. We explore a diverse set of model architectures and datasets, with a base setup of VGG-16 (Simonyan & Zisserman, 2015) trained on CIFAR-10 (Krizhevsky & Hinton, 2009) dataset. In particular, our experiments cover the following models and datasets.

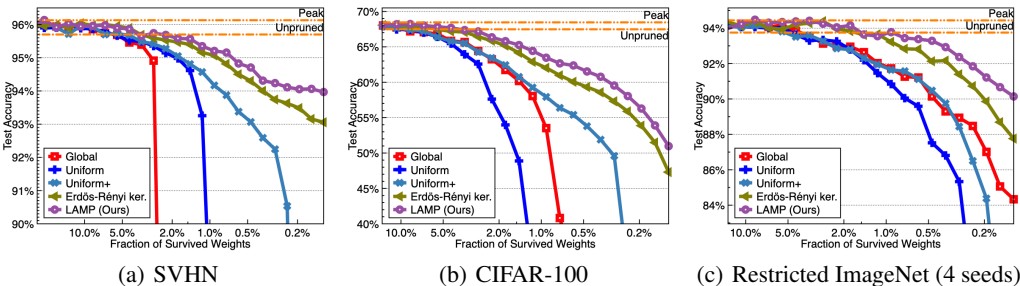

Figure 3: Sparsity-accuracy tradeoff curves of pruned models trained for SVHN and CIFAR-100 (on VGG-16) and Restricted ImageNet (on ResNet-34).

**Models.** We consider four network architectures for image classification experiments: (1) VGG-16 (Simonyan & Zisserman, 2015) adapted for CIFAR-10 to have batch normalization layers and one fully-connected layer (as used in Liu et al. (2019); Frankle & Carbin (2019)); (2) ResNet-20/34 (He et al., 2016); (3) DenseNet-121 (Huang et al., 2017); (4) EfficientNet-B0 (Tan & Le, 2019). For all four models, we prune the weight tensors for fully-connected and convolutional units. Biases and batch normalization layers are kept unpruned.

**Datasets.** We consider the following datasets; CIFAR-10/100 (Krizhevsky & Hinton, 2009), SVHN (Netzer et al., 2011), and Restricted ImageNet (Tsipras et al., 2019). All datasets except for Restricted ImageNet are used for training VGG-16; Restricted ImageNet is used for training ResNet-34.

**Other details.** Detailed experimental setup is given in Appendix A.

In Fig. 2, we provide sparsity-accuracy tradeoff curves for four different model architectures trained on CIFAR-10 dataset. The first observation is that LAMP achieves the best tradeoff; in all four models, LAMP consistently outperforms baseline methods. We also observe that Erdős-Rényi kernel method also outperforms other baselines in VGG-16, ResNet-20, and EfficientNet-B0, but fails to do so on DenseNet-121. Furthermore, the gap between LAMP and the Erdős-Rényi kernel method seems to widen as the model architecture gets more complicated; the gap between two methods is notable especially in EfficientNet-B0, where mobile inverted bottleneck convolutional layers replace traditional convolutional modules. In particular, LAMP achieves $88.1\%$ test accuracy when only $1.44\%$ of all weights survive, while Erdős-Rényi kernel achieves $77.8\%$. Lastly, we observe that the heuristic of Gale et al. (2019) seems to provide an improvement over the Uniform MP.

In Fig. 3, we present the tradeoff curves for three additional datasets: SVHN, CIFAR-100, and Restricted ImageNet. Similar to Fig. 2, we observe that LAMP outperforms all other baselines and Erdős-Rényi kernel remains to be the most competitive baseline.

## 4.2 Ablations: One-shot pruning, Weight rewinding, and SNIP

Modern magnitude-based pruning algorithms are often used in combination with customized pruning schedules (e.g., Zhu & Gupta (2018)) or weight rewinding (e.g., Frankle & Carbin (2019); Renda et al. (2020)). As a sanity check that LAMP perform reliably well along with such techniques, we conduct following additional experiments.

- **One-shot pruning.** As an extreme case of pruning schedule, we test the scheme where we only run a single training-pruning-retraining cycle, instead of iterating multiple cycles. We test the one-shot pruning on VGG-16 trained with CIFAR-10 dataset.

- **Weight rewinding.** After pruning, we rewind the remaining weights to their values in the early epoch, as in the "lottery ticket" experiments by Frankle & Carbin (2019). We perform iterative magnitude-based pruning (IMP) on VGG-16, using the warm-up step and the training schedule described in Frankle et al. (2020).

- **SNIP.** As an additional experiment, we test whether LAMP can provide a general-purpose layerwise sparsity for pruning schemes other than MP. We test under the "pruning at initialization" setup with SNIP scores (Lee et al., 2019). Baselines are similarly modified to use the SNIP score. We use Conv-6 model on CIFAR-10 dataset (see Frankle & Carbin (2019) for more details of the model) with a batch size of 128.

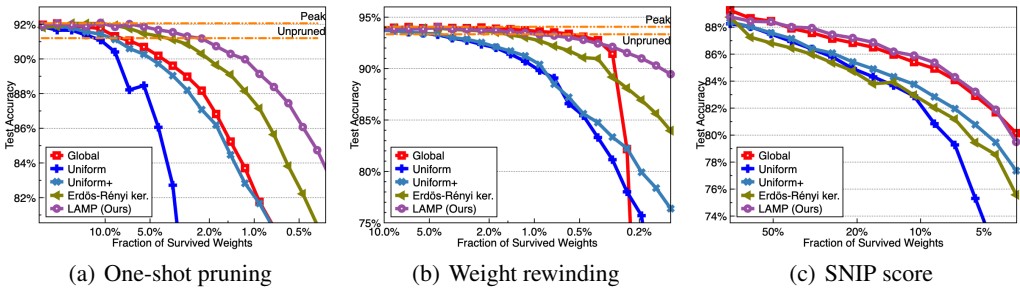

|     |     |     |
| :-: | :-: | :-: |
| (a) One-shot pruning | (b) Weight rewinding | (c) SNIP score |

Figure 4: Sparsity-accuracy tradeoff curves under one-shot pruning, weight rewinding, and the SNIP setup. One-shot pruning and the weight-rewinding experiments are done on VGG-16 trained on CIFAR-10 dataset. SNIP experiment is performed on Conv-6 trained on CIFAR-10.

Again, other experimental details are given at the Appendix A.

Results for all three experiments are given in Fig. 4. On one-shot pruning, we confirm that LAMP comfortably leads other baselines, as in the iterative pruning case. We note that the gap between one-shot pruning and iterative pruning is quite small for LAMP; when $1.15\%$ of all prunable weights survive, iterative LAMP brings only $1.09\%$ accuracy gain over one-shot LAMP. By contrast, the gain from iteration for Uniform MP is $41.62\%$ at the same sparsity level.

On the weight-rewinding experiment, we observe that LAMP remains beneficial over the baseline methods. We also remark that the Global baseline tend to perform well in this case, even outperforming the Erdős-Rényi kernel method in the low-sparsity regime. This phenomenon seems to be connected to the observation of Zhou et al. (2019) that the initial weights and final weights of a model are highly correlated; global MP may help preserving connections with larger initial magnitudes, which play important roles in terms of signal propagation at initialization (Lee et al., 2020).

On the SNIP experiment, we observe that LAMP achieves a similar performance to the Global SNIP. Recalling that the SNIP score is designed for global pruning (Lee et al., 2019), such high performance of LAMP is unexpected. We suspect that this is because LAMP is also designed for "output distortion minimization," which shares a similar spirit with the "gradient distortion minimization."

## 5 LAYERWISE SPARSITY: GLOBAL MP, ERDŐS-RÉNYI KERNEL, AND LAMP

With the effectiveness of LAMP confirmed, we take a closer look at the layerwise sparsity discovered by LAMP. We focus on answering two questions: **Q1.** Does layerwise sparsity *distilled* from LAMP behave similarly to the heuristics constructed from experiences, e.g. the one given by Gale et al. (2019)? **Q2.** Is there any other defining characteristic of LAMP sparsity patterns, which may help us guide the design of (sparse) network architectures?

In Fig. 5, we plot the layerwise survival rates and the number of nonzero weights discovered by iteratively pruning VGG-16 (trained on CIFAR-10), by Global MP, Erdős-Rényi kernel, and LAMP. Layerwise survival rates are given for the global survival rates of $\{51.2\%, 26.2\%, 13.4\%, 6.87\%, 3.52\%\}$ (from lighter to darker). Number of nonzero weights are plotted for the pruned models with total $\{3.52\%, 1.80\%, 0.92\%, 0.47\%, 0.24\%\}$ fraction of all weights surviving.

We observe that LAMP sparsities share a similar tendency to sparsity levels given by the Erdős-Rényi kernel method. In particular, both methods tend to keep the first and layer layers of the neural network relatively dense; this property is reminiscent of the handcrafted heuristic given in Gale et al. (2019): keep the first convolutional layer unpruned, and prune at most 80% from the last fully-connected layer. While Global MP also keeps a large fraction of the last fully-connected layer unpruned, the first convolutional layer gets pruned quickly. LAMP sparsities differ from Erdős-Rényi kernel sparsities in two senses.

- Although LAMP demonstrates its tendency to keep the first and the last layers relatively unpruned, this tendency is softer. When $3.52\%$ of weights survive, LAMP keeps $\sim 79\%$ and $\sim 62\%$ of weights unpruned from the first and last layers (respectively), while Erdős-Rényi kernel does not prune any weight from those two layers.

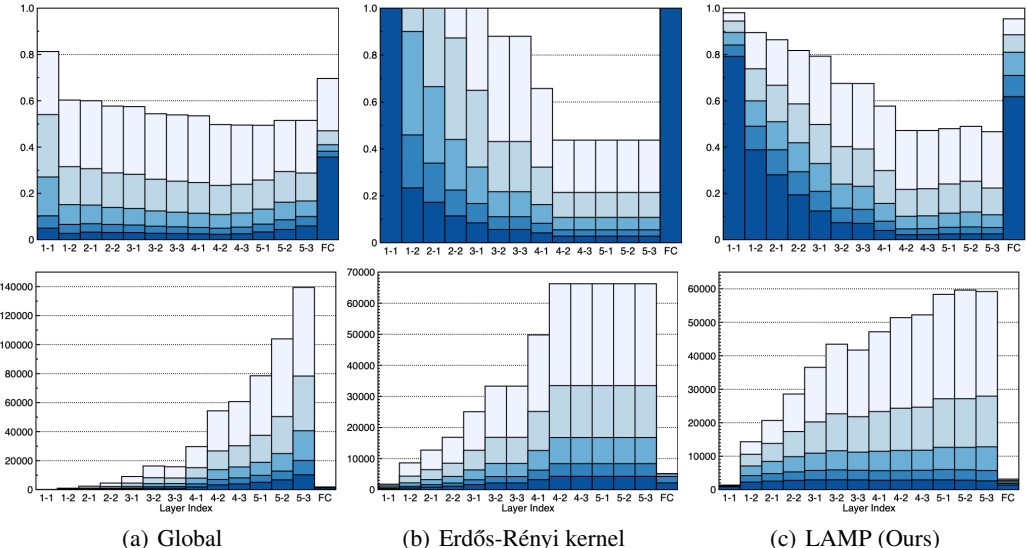

(a) Global        (b) Erdős-Rényi kernel        (c) LAMP (Ours)

Figure 5: Layerwise statistics of VGG-16 iteratively pruned on CIFAR-10. **Top:** Layerwise survival rate for models with $\{51.2\%, 26.2\%, 13.4\%, 6.87\%, 3.52\%\}$ weights surviving. **Bottom:** Number of nonzero weights for models with $\{3.52\%, 1.80\%, 0.92\%, 0.47\%, 0.24\%\}$ weights surviving.

- LAMP tends to keep the number of nonzero weights relatively uniform throughout the layers at extreme sparsity levels (indeed, the first observation can be understood as a consequence of the second observation). In contrast, Erdős-Rényi kernel method keeps the relative ratio constant, regardless of the global sparsity level.

Following the second observation, we conjecture that having a similar number of nonzero connections in each layer may be an essential condition to guarantee maximal memory capacity (see, e.g., Yun et al. (2019)), given a global sparsity constraint on the neural network. Theoretical investigations of the approximability by such sparse neural networks may be an interesting future research direction, potentially leading to a more principled and robust pruning algorithms.

As an additional remark, we note that the layerwise sparsity discovered by LAMP behaves similarly to that of AMC (He et al., 2018), which uses a reinforcement learning agent to search over the space of all available layerwise sparsity. We provide further details in Appendix F.

## 6 CONCLUSION

In this paper, we investigate an under-explored problem on the layerwise sparsity for magnitude-based pruning scheme. The proposed method, coined LAMP (Layer-Adaptive Magnitude-based Pruning), is motivated from the $\ell_2$ distortion minimization perspective on magnitude-based pruning, and provides a consistent performance gain on a wide range of models and datasets. Furthermore, LAMP performs reliably well when combined with one-shot pruning schedule or weight rewinding, which makes it an attractive candidate as a "go-to" layerwise sparsity for the magnitude-based pruning. Taking a deeper look at LAMP-discovered layerwise sparsities, we observe that LAMP automatically recovers the handcrafted rules for the layerwise sparsity. Furthermore, we observe that LAMP tend to keep the number of nonzero weights relatively uniform throughout the layers, as we consider more extreme sparsity levels. We conjecture that such uniformity in the number of unpruned weights may be an essential condition for a maximal expressive power of sparsified neural networks.

### ACKNOWLEDGMENTS

This work was supported in part by Institute of Information & Communications Technology Planning & Evaluation (IITP) grant funded by the Korea government (MSIT) (No.2019-0-00075, Artificial Intelligence Graduate School Program (KAIST)) in part by Samsung Advanced Institute of Technology (SAIT), and in part by the Defense Challengeable Future Technology Program of the Agency for Defense Development, Republic of Korea.

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

## A   EXPERIMENTAL SETUPS

For any implementational details not given in this section, we refer to the code at:

https://github.com/jaeho-lee/layer-adaptive-sparsity

**Optimizer.** With an exception of the weight rewinding experiment, we use AdamW (Loshchilov & Hutter, 2019) with learning rate 0.0003; we use vanilla Adam with learning rate 0.0003 for the weight rewinding experiment, following the setup of Frankle & Carbin (2019). For other hyperparameters, we follow the PyTorch default setup: $\vec{\beta} = (0.9, 0.999)$, wd $= 0.01$, $\varepsilon = 10^{-8}$.

**Pre-processing.** CIFAR-10/100 dataset is augmented with random crops with a padding of 4 and random horizontal flips. We normalize both training and test datasets with constants

$$(0.4914, 0.4822, 0.4465), (0.237, 0.243, 0.261).$$

SVHN training dataset is augmented with random crops with a padding of 2. We normalize both training and test datasets with constants

$$(0.4377, 0.4438, 0.4728), (0.198, 0.201, 0.197).$$

Restricted ImageNet training dataset is augmented with Random resized crop to size 224 and random horizontal flips. Restricted ImageNet test dataset is resized to 256 and then center-cropped to size 224. We normalize both training and test datasets with constants

$$(0.485, 0.456, 0.406), (0.229, 0.224, 0.225).$$

**Models.** Some of the models used for CIFAR-10 datasets are originally developed for ImageNet: VGG-16, DenseNet-121, EfficientNet-B0. The models are adapted for CIFAR-10/100 datasets by modifying only the average pooling and the (first) fully-connected layer (and not convolutional layers) to fit the $32 \times 32$ resolution of the input image.

Table 1: Optimization details.

| Dataset | Model | Initial training iter. | Re-training iter. | Batch size |
|---|---|---|---|---|
| SVHN | VGG-16 | 40000 | 30000 | 100 |
| CIFAR-10 | {VGG-16, EfficientNet-B0} | 50000 | 40000 | 100 |
| CIFAR-10 | DenseNet-121 | 80000 | 60000 | 100 |
| CIFAR-100 | VGG-16 | 60000 | 50000 | 100 |
| Restricted ImageNet | ResNet-34 | 80000 | 80000 | 128 |
| CIFAR-10 | Conv-6 (SNIP) | 50000 | 40000 | 128 |

## B  COMPUTATIONAL AND IMPLEMENTATIONAL ASPECTS OF LAMP

In this section, we discuss the computational complexity of LAMP and describe how LAMP can be implemented. We consider a depth-$d$ neural network, with $n_i$ number of connections in $i$-th layer. We also let $n = \sum_{i=1}^{d} n_i$.

The global MP is comprised of two steps:

- **Step 1.** Pool and sort the weight magnitudes of all connections, taking $\mathcal{O}(n \log n)$ operations.
- **Step 2.** Assign 0s to connections with smallest weights until the desired sparsity level is achieved, taking $\mathcal{O}(n)$ computations.

The total computational cost is of order $\mathcal{O}(n \log n)$. Similarly, we see that performing MP with any pre-determined sparsity levels incurs $\mathcal{O}(\sum_{i=1}^{d} n_i \log n_i)$ computations.

LAMP, on the other hand, can be done in four steps.

- **Step 1.** Weight magnitudes are squared and sorted for each layer, which can be done with a computation cost of $\mathcal{O}(\sum_{i=1}^{d} n_i \log n_i)$.
- **Step 2.** LAMP score denominators $\sum_{v \geq u} W_i^2[v]$ for each layer is computed by summing-and-storing the squared weight magnitudes according to a descending order; takes $\mathcal{O}(n)$ computations.
- **Step 3.** The LAMP score is computed by dividing squared weight magnitudes by the denominators, using $\mathcal{O}(n)$ steps.
- **Step 4.** We sort and prune as in global MP, taking $\mathcal{O}(n \log n)$ steps.[7]

We observe that step 4 is the dominant term, which is shared by the global MP. The first three steps can be easily implemented in PyTorch as follows.

```
def lamp_score(weight):
    normalizer = weight.norm() ** 2

    sorted_weight, sorted_idx = weight.abs().view(-1).sort(descending=False)

    weight_cumsum_temp = (sorted_weight ** 2).cumsum(dim=0)
    weight_cumsum = torch.zeros(weight_cumsum_temp.shape)
    weight_cumsum[1:] = weight_cumsum_temp[:len(weight_cumsum_temp) - 1]

    sorted_weight /= (normalizer - weight_square_cumsum).sqrt()

    score = torch.zeros(weight_cumsum.shape)
    score[sorted_idx] = sorted_weight

    score = score.view(weight.shape)

    return score
```

---

[7]This cost can be further reduced, recalling that the LAMP scores in each layer are already sorted. All that remains is a merging step.

## C   DERIVATION OF INEQUALITY (8)

In this section, we prove the following inequality.

$$\sup_{\|x\|_2 \leq 1} \|f(x; W^{(1:d)}) - f(x; W^{(1:i-1)}, \widetilde{W}^{(i)}, W^{(i+1:d)})\|_2 \leq \|W^{(i)} - \widetilde{W}^{(i)}\|_F \cdot \prod_{j \neq i} \|W^{(j)}\|_F. \tag{9}$$

This inequality is a simplified and modified version of what is popularly known as "peeling" procedure in the generalization literature (e.g., (Neyshabur et al., 2015)), and we present the proof only for the sake of completeness. We begin by peeling the outermost layer as

$$\|f(x; W^{(1:d)}) - f(x; W^{(1:i-1)}, \widetilde{W}^{(i)}, W^{(i+1:d)})\|_2 \tag{10}$$

$$= \left\| W^{(d)} \left( \sigma(f(x; W^{(1:d-1)})) - \sigma(f(x; W^{(1:i-1)}, \widetilde{W}^{(i)}, W^{(i+1:d-1)})) \right) \right\|_2 \tag{11}$$

$$\leq \|W^{(d)}\|_F \cdot \left\| \sigma(f(x; W^{(1:d-1)})) - \sigma(f(x; W^{(1:i-1)}, \widetilde{W}^{(i)}, W^{(i+1:d-1)})) \right\|_2 \tag{12}$$

$$\leq \|W^{(d)}\|_F \cdot \left\| f(x; W^{(1:d-1)}) - f(x; W^{(1:i-1)}, \widetilde{W}^{(i)}, W^{(i+1:d-1)}) \right\|_2, \tag{13}$$

where we have used Cauchy-Schwarz inequality for the first inequality, and the 1-Lipschitzness of ReLU activation with respect to $\ell_2$ norm. We keep on peeling until we get

$$\|f(x; W^{(1:d)}) - f(x; W^{(1:i-1)}, \widetilde{W}^{(i)}, W^{(i+1:d)})\|_2 \tag{14}$$

$$\leq \left( \prod_{j > i} \|W^{(j)}\|_F \right) \cdot \|f(x; W^{(1:i)}) - f(x; W^{(1:i-1)}, \widetilde{W}^{(i)})\|_2. \tag{15}$$

The second multiplicative term on the right-hand side requires a slightly different treatment. Via Cauchy-Schwarz, we get

$$\|f(x; W^{(1:i)}) - f(x; W^{(1:i-1)}, \widetilde{W}^{(i)})\|_2 = \left\| \left( W^{(i)} - \widetilde{W}^{(i)} \right) \sigma(f(x; W^{(1:i-1)})) \right\|_2 \tag{16}$$

$$\leq \|W^{(i)} - \tilde{W}^{(i)}\|_F \cdot \|\sigma(f(x; W^{(1:i-1)}))\|_2. \tag{17}$$

The activation functions from this point require zero-in-zero-out (ZIZO; $\sigma(\mathbf{0}) = \mathbf{0}$) property to be peeled, in addition to the Lipschitzness. Indeed, we can proceed as

$$\|\sigma(f(x; W^{(1:i-1)}))\|_2 = \|\sigma(f(x; W^{(1:i-1)})) - \sigma(\mathbf{0})\|_2 \leq \|f(x; W^{(1:i-1)}) - \mathbf{0}\|_2 \tag{18}$$

$$= \|f(x; W^{(1:i-1)})\|_2. \tag{19}$$

Iteratively applying the Cauchy-Schwarz inequality and the step Eq. (19), we arrive at Eq. (9).

# D  EXPERIMENTAL RESULTS ON LANGUAGE MODELING

In this section, we apply the proposed LAMP and baseline layerwise sparsities on language modeling tasks. We note, however, that it is still unclear whether magnitude-based pruning approaches achieve state-of-the-art results for pruning natural language processing tasks (see, e.g., Sanh et al. (2020), for more discussions). In particular, NLP model architectures (e.g., embedding layers) and learning pipelines (e.g., more prevalent use of pre-trained models) add more complications to model pruning, obscuring how pruning algorithms should be fairly compared. We also note that we do not test Uniform+ baseline for this task, as the heuristic is proposed explicitly for image classification models, not language models (Gale et al., 2019).

**Model.** We consider the simplified version of Transformer-XL (Dai et al., 2018) architecture, which has six self-attention layers (original has 16) and removed some regularizing heuristics, such as exponential moving average and cosine annealing. Similar to the setting of Sanh et al. (2020), we focus on pruning self-attention layers of the model (approximately 7.57M parameters).

**Optimizer.** We use AdamW (Loshchilov & Hutter, 2019) with learning rate 0.0003, following the PyTorch default setup for other hyperparameters: $\vec{\beta} = (0.9, 0.999)$, wd $= 0.01$, $\varepsilon = 10^{-8}$. We use batch size 20 and maximum sequence length 70. We train the initial unpruned model for 200,000 iterations and re-train the model after pruning for 50,000 iterations.

**Datasets.** We use Penn Treebank (Marcus et al., 1993) and WikiText-2 (Merity et al., 2016) datasets.

We provide the experimental results in Tables 2 and 3. We observe that LAMP consistently achieves the near-best performance among all considered methods. We note, however, that the gain is marginal. We suspect that this marginal-ity is due to the fact that *widths* of the Transformer layers are relatively uniform compared to image classification models. We do not observe any notable pattern among the baseline methods.

Table 2: Test perplexities of Transformer-XL iteratively pruned and trained on Penn Treebank. Bold denotes the pruning scheme with accuracy within one standard deviation from the highest.

| Surv. weights | 26.2% | 21.0% | 16,8% | 13.4% | 10.8% | 8.59% | 6.87% | 5.50% | 4.40% | 3.52% |
|---|---|---|---|---|---|---|---|---|---|---|
| Global | **71.38**±0.39 | **71.78**±0.34 | **73.23**±0.38 | **73.67**±0.10 | **76.23**±0.43 | **78.21**±0.31 | 80.92±0.10 | 83.98±0.36 | 88.62±0.25 | 92.30±0.54 |
| Uniform | **71.57**±0.46 | **71.74**±0.39 | **73.27**±0.36 | 74.18±0.34 | 76.98±0.41 | 79.25±0.29 | 81.79±0.13 | 84.95±0.35 | 89.41±0.39 | 92.44±0.69 |
| Erdős-Rényi ker. | **71.53**±0.33 | **71.94**±0.20 | 73.56±0.31 | 74.43±0.20 | 77.15±0.33 | 79.38±0.26 | 82.03±0.13 | 85.01±0.38 | 89.34±0.31 | 92.75±0.55 |
| LAMP (Ours) | **71.51**±0.38 | **71.79**±0.24 | **73.11**±0.37 | **73.74**±0.21 | **76.23**±0.27 | **78.27**±0.18 | **80.73**±0.09 | **83.60**±0.35 | **88.06**±0.24 | **91.24**±0.38 |

Table 3: Test perplexities of Transformer-XL iteratively pruned and trained on WT-2. Bold denotes the pruning scheme with accuracy within one standard deviation from the highest.

| Surv. weights | 26.2% | 21.0% | 16,8% | 13.4% | 10.8% | 8.59% | 6.87% | 5.50% | 4.40% | 3.52% |
|---|---|---|---|---|---|---|---|---|---|---|
| Global | **87.62**±0.28 | **90.43**±0.58 | **91.35**±0.21 | **94.65**±0.30 | **96.34**±0.35 | 100.07±0.48 | 103.56±0.48 | 107.95±0.74 | 112.20±0.34 | 117.03±0.63 |
| Uniform | **87.88**±0.26 | **90.46**±0.69 | 91.58±0.29 | 95.01±0.21 | 96.72±0.79 | 100.83±0.32 | 104.25±0.28 | 108.31±0.53 | 112.71±0.25 | 117.01±0.50 |
| Erdős-Rényi ker. | 88.01±0.23 | **90.62**±0.64 | 91.99±0.28 | 95.40±0.36 | 97.13±0.72 | 100.83±0.38 | 104.49±0.51 | 108.51±0.90 | 112.56±0.47 | 116.69±0.34 |
| LAMP (Ours) | **87.73**±0.26 | **90.76**±0.54 | **91.47**±0.33 | **94.58**±0.30 | **96.10**±0.53 | **99.62**±0.19 | **103.01**±0.22 | **106.60**±0.46 | **110.87**±0.63 | **115.31**±0.64 |

# E  DETAILED EXPERIMENTAL RESULTS (WITH STANDARD DEVIATIONS)

Table 4: Test accuracies of VGG-16 iteratively pruned and trained on CIFAR-10. Bold denotes the pruning scheme with accuracy within one standard deviation from the highest.

| Surv. weights | 6.87% | 4.40% | 2.81% | 1.80% | 1.15% | 0.74% | 0.47% | 0.30% | 0.19% | 0.12% |
|---|---|---|---|---|---|---|---|---|---|---|
| Global | $91.30_{\pm0.23}$ | $90.80_{\pm0.23}$ | $89.28_{\pm0.30}$ | $85.55_{\pm2.38}$ | $81.56_{\pm3.73}$ | $54.58_{\pm22.07}$ | $41.91_{\pm18.63}$ | $31.93_{\pm21.24}$ | $21.87_{\pm21.21}$ | $11.72_{\pm2.29}$ |
| Uniform | $91.47_{\pm0.19}$ | $90.78_{\pm0.12}$ | $88.61_{\pm1.09}$ | $84.17_{\pm4.46}$ | $55.68_{\pm12.20}$ | $38.51_{\pm24.83}$ | $26.41_{\pm8.83}$ | $16.75_{\pm6.62}$ | $11.58_{\pm2.79}$ | $9.95_{\pm0.28}$ |
| Uniform+ | $91.54_{\pm0.19}$ | $91.20_{\pm0.28}$ | $90.16_{\pm0.12}$ | $89.44_{\pm0.15}$ | $87.85_{\pm0.26}$ | $86.53_{\pm0.05}$ | $84.84_{\pm0.41}$ | $82.41_{\pm0.44}$ | $74.54_{\pm5.41}$ | $24.46_{\pm14.98}$ |
| Erdős-Rényi ker. | $\mathbf{92.34}_{\pm0.25}$ | $\mathbf{91.99}_{\pm0.14}$ | $\mathbf{91.66}_{\pm0.29}$ | $91.15_{\pm0.24}$ | $90.55_{\pm0.19}$ | $89.51_{\pm0.50}$ | $88.21_{\pm0.38}$ | $86.73_{\pm0.25}$ | $84.85_{\pm0.28}$ | $81.50_{\pm0.42}$ |
| LAMP (Ours) | $\mathbf{92.24}_{\pm0.24}$ | $\mathbf{92.06}_{\pm0.21}$ | $\mathbf{91.71}_{\pm0.30}$ | $\mathbf{91.66}_{\pm0.21}$ | $\mathbf{91.07}_{\pm0.40}$ | $\mathbf{90.49}_{\pm0.21}$ | $\mathbf{89.64}_{\pm0.32}$ | $\mathbf{88.75}_{\pm0.29}$ | $\mathbf{87.07}_{\pm0.37}$ | $\mathbf{84.90}_{\pm0.59}$ |

Table 5: Test accuracies of ResNet-20 iteratively pruned and trained on CIFAR-10. Bold denotes the pruning scheme with accuracy within one standard deviation from the highest.

| Surv. weights | 20.97% | 13.42% | 8.59% | 5.50% | 3.52% | 2.25% | 1.44% | 0.92% | 0.59% | 0.38% |
|---|---|---|---|---|---|---|---|---|---|---|
| Global | $87.48_{\pm0.28}$ | $86.97_{\pm0.39}$ | $86.29_{\pm0.23}$ | $85.02_{\pm0.35}$ | $83.15_{\pm0.33}$ | $80.52_{\pm0.36}$ | $76.28_{\pm0.50}$ | $70.69_{\pm0.32}$ | $47.47_{\pm11.13}$ | $12.02_{\pm3.77}$ |
| Uniform | $87.24_{\pm0.28}$ | $86.70_{\pm0.20}$ | $86.09_{\pm0.21}$ | $84.53_{\pm0.34}$ | $82.05_{\pm0.23}$ | $77.19_{\pm2.36}$ | $64.24_{\pm14.50}$ | $47.97_{\pm2.91}$ | $20.45_{\pm8.73}$ | $13.35_{\pm2.66}$ |
| Uniform+ | $87.30_{\pm0.29}$ | $87.00_{\pm0.23}$ | $86.27_{\pm0.13}$ | $85.00_{\pm0.15}$ | $83.23_{\pm0.19}$ | $80.40_{\pm0.31}$ | $76.40_{\pm0.45}$ | $69.31_{\pm1.27}$ | $52.06_{\pm2.53}$ | $20.19_{\pm8.44}$ |
| Erdős-Rényi ker. | $\mathbf{87.63}_{\pm0.10}$ | $\mathbf{87.49}_{\pm0.31}$ | $\mathbf{86.83}_{\pm0.27}$ | $\mathbf{85.84}_{\pm0.23}$ | $\mathbf{84.08}_{\pm0.34}$ | $\mathbf{81.76}_{\pm0.37}$ | $\mathbf{78.70}_{\pm0.25}$ | $\mathbf{74.40}_{\pm0.52}$ | $\mathbf{66.42}_{\pm1.46}$ | $\mathbf{50.90}_{\pm3.28}$ |
| LAMP (Ours) | $87.54_{\pm0.37}$ | $87.12_{\pm0.11}$ | $\mathbf{86.56}_{\pm0.21}$ | $\mathbf{85.64}_{\pm0.26}$ | $\mathbf{84.18}_{\pm0.23}$ | $\mathbf{81.56}_{\pm0.26}$ | $\mathbf{78.63}_{\pm0.50}$ | $\mathbf{74.20}_{\pm0.81}$ | $\mathbf{67.01}_{\pm0.92}$ | $\mathbf{51.24}_{\pm8.48}$ |

Table 6: Test accuracies of DenseNet-121 iteratively pruned and trained on CIFAR-10. Bold denotes the pruning scheme with accuracy within one standard deviation from the highest.

| Surv. weights | 5.50% | 4.40% | 3.52% | 2.82% | 2.25% | 1.80% | 1.44% | 1.15% | 0.92% | 0.74% |
|---|---|---|---|---|---|---|---|---|---|---|
| Global | $90.16_{\pm0.36}$ | $89.52_{\pm0.36}$ | $88.83_{\pm0.44}$ | $88.00_{\pm0.43}$ | $86.85_{\pm0.47}$ | $85.32_{\pm0.58}$ | $77.68_{\pm8.01}$ | $45.30_{\pm27.75}$ | $49.65_{\pm21.88}$ | $20.96_{\pm19.49}$ |
| Uniform | $90.24_{\pm0.11}$ | $89.50_{\pm0.13}$ | $88.44_{\pm0.19}$ | $87.94_{\pm0.45}$ | $86.83_{\pm0.41}$ | $85.00_{\pm1.54}$ | $82.16_{\pm3.37}$ | $70.13_{\pm13.69}$ | $66.46_{\pm18.72}$ | $48.71_{\pm21.42}$ |
| Uniform+ | $90.25_{\pm0.18}$ | $89.70_{\pm0.20}$ | $89.03_{\pm0.17}$ | $88.22_{\pm0.27}$ | $87.40_{\pm0.41}$ | $86.26_{\pm0.44}$ | $84.55_{\pm0.45}$ | $81.87_{\pm0.77}$ | $69.25_{\pm19.28}$ | $58.91_{\pm8.26}$ |
| Erdős-Rényi ker. | $90.21_{\pm0.29}$ | $89.79_{\pm0.29}$ | $88.92_{\pm0.07}$ | $88.20_{\pm0.23}$ | $87.25_{\pm0.36}$ | $86.22_{\pm0.30}$ | $84.11_{\pm0.50}$ | $81.82_{\pm0.90}$ | $59.06_{\pm25.61}$ | $59.07_{\pm25.70}$ |
| LAMP (Ours) | $\mathbf{90.89}_{\pm0.17}$ | $\mathbf{90.11}_{\pm0.13}$ | $\mathbf{89.72}_{\pm0.26}$ | $\mathbf{89.12}_{\pm0.35}$ | $\mathbf{88.39}_{\pm0.26}$ | $\mathbf{87.75}_{\pm0.18}$ | $\mathbf{86.53}_{\pm0.33}$ | $\mathbf{85.13}_{\pm0.31}$ | $\mathbf{82.92}_{\pm0.66}$ | $\mathbf{79.23}_{\pm1.29}$ |

Table 7: Test accuracies of EfficientNet-B0 iteratively pruned and trained on CIFAR-10. Bold denotes the pruning scheme with accuracy within one standard deviation from the highest.

| Surv. weights | 41.0% | 26.2% | 16.8% | 10.7% | 6.87% | 4.40% | 2.82% | 1.80% | 1.15% | 0.74% |
|---|---|---|---|---|---|---|---|---|---|---|
| Global | $\mathbf{89.66}_{\pm0.21}$ | $89.55_{\pm0.30}$ | $88.80_{\pm0.32}$ | $87.64_{\pm0.35}$ | $84.36_{\pm0.59}$ | $79.25_{\pm0.67}$ | $11.09_{\pm2.14}$ | $10.62_{\pm1.38}$ | $10.00_{\pm0.00}$ | $10.00_{\pm0.00}$ |
| Uniform | $88.99_{\pm0.31}$ | $88.26_{\pm0.32}$ | $86.48_{\pm0.38}$ | $83.40_{\pm0.38}$ | $23.65_{\pm28.16}$ | $10.83_{\pm0.83}$ | $10.00_{\pm0.00}$ | $10.00_{\pm0.00}$ | $10.00_{\pm0.00}$ | $10.00_{\pm0.00}$ |
| Uniform+ | $89.18_{\pm0.35}$ | $88.03_{\pm0.19}$ | $86.71_{\pm0.40}$ | $84.16_{\pm0.65}$ | $36.64_{\pm35.77}$ | $10.45_{\pm0.67}$ | $10.00_{\pm0.00}$ | $10.19_{\pm1.62}$ | $10.00_{\pm0.00}$ | $10.00_{\pm0.00}$ |
| Erdős-Rényi ker. | $\mathbf{89.54}_{\pm0.17}$ | $\mathbf{90.09}_{\pm0.09}$ | $\mathbf{90.01}_{\pm0.31}$ | $89.62_{\pm0.26}$ | $88.82_{\pm0.39}$ | $87.08_{\pm0.32}$ | $84.72_{\pm0.24}$ | $81.53_{\pm0.44}$ | $51.31_{\pm25.77}$ | $13.40_{\pm3.04}$ |
| LAMP (Ours) | $\mathbf{89.52}_{\pm0.51}$ | $\mathbf{89.95}_{\pm0.20}$ | $\mathbf{89.97}_{\pm0.23}$ | $\mathbf{90.21}_{\pm0.28}$ | $\mathbf{89.91}_{\pm0.14}$ | $\mathbf{89.79}_{\pm0.33}$ | $\mathbf{89.30}_{\pm0.08}$ | $\mathbf{88.51}_{\pm0.37}$ | $\mathbf{86.79}_{\pm0.73}$ | $\mathbf{65.76}_{\pm20.07}$ |

Table 8: Test accuracies of VGG-16 iteratively pruned and trained on SVHN. Bold denotes the pruning scheme with accuracy within one standard deviation from the highest.

| Surv. weights | 6.87% | 4.40% | 2.82% | 1.80% | 1.15% | 0.74% | 0.47% | 0.30% | 0.19% | 0.12% |
|---|---|---|---|---|---|---|---|---|---|---|
| Global | $95.75_{\pm0.17}$ | $95.46_{\pm0.10}$ | $94.91_{\pm0.48}$ | $61.06_{\pm30.93}$ | $65.06_{\pm11.01}$ | $49.51_{\pm24.23}$ | $26.83_{\pm19.39}$ | $15.37_{\pm5.95}$ | $7.79_{\pm1.60}$ | $14.72_{\pm3.65}$ |
| Uniform | $95.78_{\pm0.16}$ | $95.52_{\pm0.10}$ | $95.35_{\pm0.14}$ | $94.98_{\pm0.19}$ | $93.26_{\pm1.42}$ | $64.32_{\pm15.03}$ | $29.54_{\pm10.21}$ | $19.63_{\pm7.54}$ | $13.73_{\pm6.97}$ | $17.94_{\pm4.85}$ |
| Uniform+ | $95.70_{\pm0.13}$ | $95.68_{\pm0.17}$ | $95.42_{\pm0.11}$ | $95.05_{\pm0.15}$ | $94.57_{\pm0.16}$ | $93.87_{\pm0.10}$ | $93.07_{\pm0.16}$ | $92.25_{\pm0.25}$ | $74.88_{\pm25.75}$ | $17.67_{\pm14.33}$ |
| Erdős-Rényi ker. | $\mathbf{96.00}_{\pm0.09}$ | $\mathbf{95.89}_{\pm0.11}$ | $95.59_{\pm0.15}$ | $95.47_{\pm0.10}$ | $95.16_{\pm0.08}$ | $94.81_{\pm0.16}$ | $94.30_{\pm0.18}$ | $93.74_{\pm0.27}$ | $93.48_{\pm0.27}$ | $93.06_{\pm0.25}$ |
| LAMP (Ours) | $\mathbf{95.98}_{\pm0.15}$ | $\mathbf{95.87}_{\pm0.13}$ | $\mathbf{95.74}_{\pm0.10}$ | $\mathbf{95.58}_{\pm0.07}$ | $\mathbf{95.35}_{\pm0.18}$ | $\mathbf{95.16}_{\pm0.08}$ | $\mathbf{94.71}_{\pm0.18}$ | $\mathbf{94.24}_{\pm0.17}$ | $\mathbf{94.05}_{\pm0.15}$ | $\mathbf{93.97}_{\pm0.19}$ |

Table 9: Test accuracies of VGG-16 iteratively pruned and trained on CIFAR-100. Bold denotes the pruning scheme with accuracy within one standard deviation from the highest.

| Surv. weights | 13.42% | 8.59% | 5.50% | 3.52% | 2.25% | 1.44% | 0.92% | 0.59% | 0.38% | 0.24% |
|---|---|---|---|---|---|---|---|---|---|---|
| Global | $68.02_{\pm0.22}$ | $67.20_{\pm0.13}$ | $66.82_{\pm0.25}$ | $65.68_{\pm0.28}$ | $63.25_{\pm0.37}$ | $60.18_{\pm0.33}$ | $53.52_{\pm1.97}$ | $33.66_{\pm18.02}$ | $7.80_{\pm2.97}$ | $4.29_{\pm6.04}$ |
| Uniform | $67.92_{\pm0.23}$ | $67.43_{\pm0.37}$ | $66.41_{\pm0.54}$ | $63.94_{\pm0.71}$ | $57.62_{\pm2.01}$ | $48.88_{\pm4.56}$ | $32.60_{\pm14.16}$ | $14.99_{\pm8.65}$ | $3.94_{\pm3.67}$ | $3.35_{\pm2.80}$ |
| Uniform+ | $67.94_{\pm0.32}$ | $67.41_{\pm0.16}$ | $66.37_{\pm0.09}$ | $65.51_{\pm0.33}$ | $63.43_{\pm0.35}$ | $60.76_{\pm0.29}$ | $57.93_{\pm0.39}$ | $55.40_{\pm0.22}$ | $51.90_{\pm1.30}$ | $33.35_{\pm13.08}$ |
| Erdős-Rényi ker. | $\mathbf{68.07}_{\pm0.42}$ | $67.92_{\pm0.36}$ | $67.45_{\pm0.22}$ | $\mathbf{67.09}_{\pm0.31}$ | $\mathbf{65.86}_{\pm0.18}$ | $64.15_{\pm0.31}$ | $62.04_{\pm0.37}$ | $60.07_{\pm0.35}$ | $58.58_{\pm0.58}$ | $55.88_{\pm0.51}$ |
| LAMP (Ours) | $\mathbf{68.07}_{\pm0.58}$ | $\mathbf{68.22}_{\pm0.27}$ | $\mathbf{67.84}_{\pm0.31}$ | $\mathbf{67.37}_{\pm0.28}$ | $\mathbf{66.35}_{\pm0.53}$ | $\mathbf{65.25}_{\pm0.23}$ | $\mathbf{63.46}_{\pm0.28}$ | $\mathbf{62.37}_{\pm0.28}$ | $\mathbf{60.77}_{\pm0.42}$ | $\mathbf{58.05}_{\pm0.42}$ |

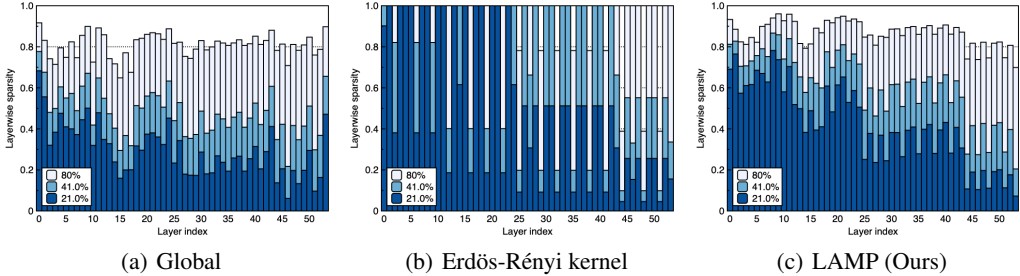

(a) Global  (b) Erdös-Rényi kernel  (c) LAMP (Ours)

Figure 6: Layerwise survival rates of one-shot pruned ResNet-50 trained on ImageNet dataset. The global survival rates are $\{80.0\%, 41.0\%, 21.0\%\}$ (from lighter to darker).

## F PEAKS AND CRESTS OF LAMP SPARSITY

In this section, we compare the layerwise sparsity discovered by LAMP (and other non-uniform baselines) to the sparsity discovered by AMC (He et al., 2018), which uses a reinforcement learning agent to explicitly search for the optimal layerwise sparsity. In particular, we focus on whether the "peaks and crests" phenomenon observed for AMC-discovered layerwise sparsity also takes place in the layerwise sparsity induced by LAMP: He et al. (2018) observe that the layerwise sparsity of ResNet-50 (trained on ImageNet) and pruned by AMC exhibits what they call peaks and crests, i.e., the sparsity oscillates between high and low periodically; see Figure 3 of He et al. (2018). The authors speculate that such phenomenon takes place, because AMC *automatically learns that* $3 \times 3$ *convolution has more redundancy than* $1 \times 1$ *convolution and can be pruned more*.

To see if LAMP also automatically discovers such "peaks and crests," we prune the ImageNet-pretrained ResNet-50 model with one-shot LAMP, global MP, and Erdös-Rényi kernel. The layerwise survival ratios are reported in Fig. 6. From the figure, we observe that LAMP also discovers different sparsity ratio for $1 \times 1$ convolution and $3 \times 3$ convolution layers; $3 \times 3$ convolutional filters are pruned more. Such pattern is more noticeable in the later layers, where more weights are pruned. In other words, LAMP discovers a layerwise sparsity similar to that of AMC, even without training a reinforcement learning agent. We note, however, a slight discrepancy in the setting: AMC reports the sparsity from an iterative pruning setup, and the results in Fig. 6 are from a one-shot pruning setup.

In the model pruned with global MP, such pattern does not stand out. In the model pruned with Erdös-Rényi kernel method, peaks and crests are extremely evident, even more than those discovered by AMC.

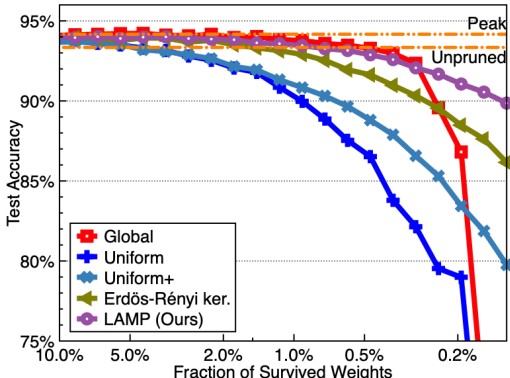

Figure 7: Performance of LAMP and baselines on VGG-16 trained on CIFAR-10 dataset, using the training schedules from Liu et al. (2019).

## G   LAMP WITH AN EXTENSIVE TRAINING SCHEDULE

In this section, we depart from the standardized training setup in the main text which uses Adam with weight decay Loshchilov & Hutter (2019), without any learning rate decay. Instead, we validate the effectiveness of LAMP using the optimizer and hyperparameters extensively tuned for the maximum performance of the model. In particular, we use configurations from Liu et al. (2019) for training VGG-16 on the CIFAR-10 dataset. The experimental result is reported in Fig. 7. Similar to the experiments appearing in the main text, we observe that LAMP continues to outperform or match the performances of the baselines.

