# OpenReview forum: "Layer-adaptive Sparsity for the Magnitude-based Pruning"
_ICLR.cc/2021/Conference — ICLR 2021 Poster_

### Official Review · AnonReviewer2 · 2020-10-27
**Well-motivated paper proposing hyperparameter-free layerwise pruning rate heuristic, though theoretical justification and evaluation could both be strengthened.**

**Rating:** 7
**Confidence:** 5

**Review:**

# Summary

The paper proposes LAMP, an importance score for unstructured pruning that incorporates layerwise statistics such that the resultant scores for each connection can be compared globally, cutting down on the hyperparameter space for magnitude pruning from the relatively standard practice of requiring hand-specified layerwise pruning rates. LAMP is motivated with a distortion analysis: LAMP is shown to be equivalent to minimizing an upper bound on the supremum of the change in model predictions of unit vectors. LAMP is compared against layerwise pruning rates obtained by standard uniform layerwise and global pruning, along with the less standard Erdos-Renyi kernel method, showing that LAMP can achieve higher accuracy for equivalent pruning rates in a specific experimental setup across several different networks.

# Strengths

- The problem the paper addresses is well-motivated: magnitude pruning rests on a poorly motivated and poorly understood set of heuristics.
- The approach proposed by the paper has several desirable properties: it is hyperparameter-free, relatively cheap to compute, and handles some corner cases of global magnitude pruning (e.g., LAMP is invariant to rescaling of a layer in a net with BatchNorm).
- Within the scope of the empirical evaluation in the paper, LAMP is demonstrated to outperform uniform and global magnitude pruning, the two most common approaches in the literature.

# Weaknesses

- Theoretical justification of LAMP: it's unclear how useful or strong the bound in Equation 7 is, especially given that the network is re-trained after pruning. These types of bounds often seem to be weak or vacuous in practice; further, there's no explicit connection between the bound and the loss, let alone the loss after re-training.
- Empirical evaluation: the training and re-training regimes used by the paper are very non-standard, raising questions about the generality of the technique and the comparison to prior work. Specifically, the choice of AdamW with a fixed learning rate and the seemingly arbitrarily chosen durations of re-training are out of line with standard practice for many of these networks; these choices mean that it is impossible to compare the accuracy/sparsity tradeoffs achieved by LAMP to the results of various other papers that claim state-of-the-art pruning results with local/global magnitude pruning.

# Overall recommendation

6: Weak accept

# Questions / suggestions for authors

I would be willing to raise my score to a 7 with a more thorough comparison against prior work with more standard hyperparameters. Specifically, if the authors reported the results a comparison of using LAMP with the same networks and training/re-training schedule as any of [1,2,3,4], or any other specific setting that claims to be state-of-the-art, I would raise my score.

# Other comments and suggestions

- The notation of Equation 1 is a little bit confusing: I'm assuming that $W[i]$ is the $i$th element in some unrolled version of $W$, and that $W^2$ denotes element-wise squaring, but it'd be good to clarify this non-standard notation.
- It would be good to have some more details on the networks, especially since many are being used in non-standard settings (e.g., VGG and ResNet-18 on CIFAR). Specifically: how are these networks adapted to these datasets, and what are the base accuracies for each networks on each dataset?
- A deeper comparison to the layerwise sparsities (and overall pereformance) of [2] would be appreciated. For example: Section 5 of this paper notes that "LAMP tends to keep the number of nonzero weights relatively uniform throughout the layers at extreme sparsity level." Section 4.2 of [2] notes "The peaks and crests show that the RL agent automatically learns to prune $3 \times 3$ convolutional layers with larger sparsity." Is there a reason for this discrepancy between the findings of each paper?
- More details on the experimental methodologies of Figures 4(b,c), along with motivations for these experiments, would be appreciated. I am particularly confused by 4(c) -- I do not see standard SNIP scores on this plot, unless "Global" here refers to SNIP scores and not global magnitude pruning (in which case, how does standard global magnitude pruning perform here?). It would be worthwhile to include full methodological details for each of these experiments, including the training and re-training (where applicable) schemes, the specific point in training at which the importance scores are calculated, and the specific importance score used by each line on the plot.
- Minor point: I don't believe that [1] used global magnitude pruning; I believe a better citation would be [5] (though there are also other prior/contemporary examples of global heuristics, like [6] and [7])

# References used in review:

[1] Song Han, Jeff Pool, John Tran, William Dally. "Learning both weights and connections for efficient neural networks".
[2] Yihui He, Ji Lin, Zhijian Liu, Hanrui Wang, Li-Jia Li, Song Han. "AMC: AutoML for Model Compression and Acceleration on Mobile Devices"
[3] Trevor Gale, Erich Elsen, Sara Hooker. "The State of Sparsity in Deep Neural Networks"
[4] Alex Renda, Jonathan Frankle, Michael Carbin. "Comparing Rewinding and Fine-tuning in Neural Network Pruning"
[5] Jonathan Frankle, Michael Carbin. "The Lottery Ticket Hypothesis: Finding Sparse, Trainable Neural Networks"
[6] Yann Le Cun, John S. Denker, Sara A. Solla. "Optimal Brain Damage"
[7] Namhoon Lee, Thalaiyasingam Ajanthan, Philip H. S. Torr. "SNIP: Single-shot Network Pruning based on Connection Sensitivity"


# Update post author response

Thanks to the authors for the detailed response. The authors have satisfactorily responded to my main criticisms of the paper (primarily about the non-standard evaluation regime, and secondarily about the motivation and strength of the theoretical results), so I’m raising my score to a 7, though I do think the paper would be further improved with references to Appendix G in the main body of the paper.

---

> ### Author Response · Authors · 2020-11-22
> **Response to R2: Part 1**
>
> Response to R2.
>
> Dear R2,
>
> Thank you for a (very detailed) review on our manuscript, and providing various suggestions.
>
> ---
> __Non-standard training recipe.__
> Following your suggestion, we ran the VGG-16 + CIFAR-10 again, using the exact experimental setup of [Frankle et al. (2020)](https://arxiv.org/abs/1912.05671), which in turn inherits the settings of [ICLR 2019 paper by Liu et al.](https://arxiv.org/abs/1810.05270); we selected this setup as it seems to be the state-of-the-art for VGG-16+CIFAR-10 setup (with full hyperparameters available), which is the base experimental setup for our paper but has not been discussed in [1,2,3,4].
>
> The experimental result is added to the **Appendix G**. We averaged over five independent trials for reproducibility, similarly to other experimental results in the paper.
>
> _Observations_: Similarly to the original setup, we observe that (1) LAMP outperforms/matches the performance of the strongest baselines, which are global MP (in the low-sparsity regime) and ERK (in the high-sparsity regime), and (2) ERK > Uniform+ > Uniform holds. One remark is that the performance of global MP tends to be very sensitive to the training recipe, compared to other baselines. LAMP, on the other hand, performs more robustly well under different training setups and achieves a better performance, especially in the high-sparsity regime.
>
>
>
>
> __Theoretical justification of LAMP.__
> As the reviewer pointed out, the theoretically motivated design of the LAMP score does not lead to a theoretical performance guarantee of the pruned-retrained model. We would like to clarify, however, that the primary role of the provided inequality (eq. 7) is to motivate the LAMP score and deliver the overall insight. In a sense, this is similar to how [OBD](https://papers.nips.cc/paper/1989/file/6c9882bbac1c7093bd25041881277658-Paper.pdf) is designed; LeCun et al. motivates the score from the Taylor series decomposition of loss, but the OBD score itself cannot be directly related to the actual loss (even before the retraining), as (1) multiple connections are pruned simultaneously and (2) there should exist higher-order terms as well. The difference from OBD is that the LAMP score is designed having the *output distortion* in mind, which can be thought of as an “$\ell_2$ loss with respect to the worst-case input.”
>
> We clarified these points in **Section 3.1** of the revised draft.
>
> ---
> __Comparison with AMC.__
> We deeply appreciate the pointer to the [AMC paper.](https://arxiv.org/abs/1802.03494), and the suggestion for more discussions. We dug a little deeper into the apparent discrepancy between the AMC-discovered layerwise sparsity and LAMP-discovered sparsity.
>
> We believe that the discrepancy is due to the discrepancy of model architectures under consideration. According to the AMC paper, “peaks and crests” take place for ResNet-50 because “the reinforcement learning agent automatically learns that 3x3 convolution has more redundancy than 1x1 convolutions.” The AMC sparsities of Plain-20 do not exhibit such clearly visible peaks-and-crests. Similar to Plain-20, we reported the layerwise sparsities for VGG-16 composed of 3x3 convolutions.
>
> Can LAMP recover “peaks and crests” for the mixture-of-1x1-and-3x3 models like ResNet-50? To answer this question, we took a PyTorch pretrained ResNet-50 and performed one-shot pruning with LAMP (without an inference/training step). We report the results on the newly added Appendix F. In Figure 6, we observe that the layerwise sparsity of LAMP has peaks at the 1x1 sparsity. The difference from AMC-discovered sparsity is that for LAMP sparsities such behavior is clearer in the layer residual blocks, while in AMC sparsities the crests and peaks take place at earlier blocks as well.
>
> We added the discussion to **Appendix F**, which we believe that the readers will enjoy. Thanks again!

---

> ### Author Response · Authors · 2020-11-22
> **Response to R2: Part 2.**
>
> __Details on Figs. 4b-c.__
> Thank you for pointing out the need for extra details. To clarify, “Global” on Figure 4c indeed refers to “global thresholding with SNIP scores.” In fact, we simply plugged in SNIP scores in the place of magnitude scores. We revised our manuscript to provide more details to the readers in **Section 4.2**.
>
> About the motivation of Figure 4b-c: As noted in the introduction, our primary goal was to provide a layerwise sparsity scheme that could be plugged in to diverse algorithms which utilizes magnitude heuristics for pruning, regardless of auxiliary modules that constitute a full algorithm. For instance, the algorithm of [Zhu&Gupta 2017](https://arxiv.org/abs/1710.01878) can be viewed as “MP + uniform sparsity + reconnection + cubic pruning schedule.” Here, the re-connectability and cubic pruning schedule may be viewed as modules that are auxiliary to the layerwise sparsity module. In Figure 4b, we test if the LAMP sparsity works well when used jointly with the “weight rewinding module,” which seems to be a promising new component for MP algorithms as [Renda et al. (2020)](https://arxiv.org/abs/2003.02389) showed. For Figure 4c, we asked whether the LAMP sparsity can be combined with pruning scores other than the weight magnitude. While this application is not exactly theory-motivated, we interestingly observed that LAMP sparsity performs competitively with “Global SNIP,” which is more theoretically justified. Based on this observation, we would (mildly) suggest using LAMP-like sparsity determination schemes for any “not-designed-to-be-global scores” for pruning (e.g. [movement pruning](https://arxiv.org/pdf/2005.07683.pdf)).
>
> ---
> __Details on the networks.__
> Thank you for this suggestion. We have added the details to **Appendix A**, along with references from which such setting and architecture has been taken from. Also, the base accuracies are now explicitly visualized on each figure, for the ease of comparison.
>
> ---
> __Notations.__
> Following your suggestion, we revised **Section 3** to provide more detailed explanations on the notations being used.
>
> ---
> __Note on Han et al.__
> Thank you for pointing this out. We revised accordingly.
>
> ---
> Please let us know if you have any further concerns.
>
> Sincerely,
> Authors.

---

### Official Review · AnonReviewer4 · 2020-10-28
**Reviews**

**Rating:** 5
**Confidence:** 4

**Review:**

Summary
- The authors propose LAMP, a layerwise adaptive magnitude-based pruning method.  The authors conduct extensive experiments on CIFAR10/CIFAR100/SVHN and Penn Treebank to validate the method.

Pros
- somewhat novel pruning method, based on new weight score
- extensive experiments on image and language datasets

Cons
- In Equation (2), the authors point out that LAMP score is align with the order of weight squares. Therefore, one can directly prune the network based on weight squares. Why is it necessary to prune the network based on LAMP?
- The comparisons are not sufficient. The authors should compare other "pruning-retraining" methods, like network slimming [1], soft filter pruning [2], etc. Though they focus on structured pruning, the core idea can be borrowed and adapted for unstructured pruning.
- Lacking of experiments on large-scale datasets and large models, for example, on ImageNet. The performance of pruning methods  can be very sensitive and versatile when only evaluating on small datasets and models. And usually, a pruning method can be invalid when testing on ImageNet models.

---

> ### Author Response · Authors · 2020-11-12
> **Clarifications on the Role/Scope of LAMP**
>
> Dear R4,
>
> Thank you for your feedback on our manuscript.
>
> We would like to briefly answer the first two concerns you have, and straighten out some potential misunderstandings on the proposed method, LAMP.
>
> __Difference from squared magnitude.__
> As the title of our paper suggests, LAMP aims to provide a method to select the *layerwise sparsity* for the magnitude-based pruning. In other words, our goal is to do the magnitude-based pruning at each layer (to meet the global sparsity constraint), but we hope to select the layerwise sparsity level carefully to maximize the performance of the pruned models.
>
> Here, _global_ pruning (i.e., using a global threshold on the score) using the LAMP score differs from the global pruning using the square weight score. In each layer, they both do the magnitude-based pruning, but globally, they lead to different layerwise sparsity. Indeed, we show that “global pruning based on lamp” gives a better performing subnetworks (Section 4) with very different layerwise sparsity levels (Section 5) than the “global pruning based on squared magnitude.” We will clarify this in the revised manuscript.
>
> __Comparisons to channel pruning methods.__
> As we briefly mentioned above, LAMP is designed as a layerwise sparsity selection scheme for MP (which is an important subject, as Gale et al. (2019) demonstrated), that can be combined with various other algorithmic modules (gradual pruning schedule, weight rewinding) to constitute a full pruning-retraining pipeline. In this sense, we do not view that the works R4 mentioned (Network-Slimming, Soft filter pruning) should be the baseline of our method; instead, we make comparisons to other *layerwise sparsity* selection schemes for magnitude-based pruning algorithms. We will revise our introduction to emphasize this point.
>
> We hope this response will resolve some of your concerns. Please let us know if there are any unclear points remaining.
>
> Sincerely,
> Authors.

---

> ### Author Response · Authors · 2020-11-22
> **Additional response to R4**
>
> Dear R4,
>
> Thank you again for the thoughtful comments and suggestions on our manuscript. In addition to our initial response regarding the first two concerns you raised, we now respond to your third concern; thank you for your patience!
>
> __ImageNet experiments__
> As reviewer#1 also noted, ImageNet experiments are quite computationally expensive; each pruning-retraining iteration takes $~1.8$ days with the computational resource that we have access to. Thus, to best respond to the reviewer’s question, we conducted _one-shot pruning_ on ResNet-18 + ImageNet setup, instead of the iterative pruning: we used the sparsity 10.74%. We will try to have the iterative pruning result ready for the final version.
>
> The experimental result is as follows: (figures denote the error rate.)
> unpruned [test@1 30.46] [test@5 10.84]
>
> lamp [test@1 41.40] [test@5 18.09]
> erk [test@1 41.64] [test@5 18.41]
> glob [test@1 42.34] [test@5 18.98]
> unif [test@1 45.58] [test@5 21.46]
>
> Finally, we note that we already report the results on the [Restricted ImageNet](https://arxiv.org/abs/1805.12152) dataset, which contains ~268k images of ImageNet resolution.
>
> ---
> Please let us know if you have any further feedback!
>
> Sincerely,
> Authors.

---

### Official Review · AnonReviewer3 · 2020-10-28
**Strong experimental results and an elegant technique- more experimental diversity would be beneficial**

**Rating:** 8
**Confidence:** 3

**Review:**

This paper presents a novel technique  (layer-adaptive magnitude based pruning, or LAMP) for pruning neural network weights (pruning can be beneficial in terms of overfitting prevention as well as other practical considerations).  LAMP evaluates weights in each layer in terms of the ratio of the magnitude of the weight to the sum of magnitudes of all surviving weights in the layer. The weight which evaluates as least important across all layers is pruned and then the process is repeated until the desired sparsity is achieved. The method is motivated theoretically as minimizing the distortion in the input/output mapping implemented by the weights of the layer. Experimental results on several benchmarks are presented.

Pros:

The experimental results are strong,  with LAMP consistently winning vs. competing techniques on 4 benchmarks problems.

The method is elegant, requiring no hyperparameter tuning and minimal computation.

The theoretical justification (mapping-distortion-minimization) makes a lot of sense.

Cons:

My only objection is that all of the experiments seem to be done on image datasets. It seems possible that deep learning networks applied to non-image data might not do as well under LAMP as they do for images. 'Under diverse datasets' in the abstract seems like an exaggeration.

Further comments:

I found a couple of typos.

P3 while such unstrutured pruning -> while such unstructured pruning
P5 Global ‘on every layers’ -> on every layer

---

> ### Author Response · Authors · 2020-11-22
> **Response to R3**
>
> Dear R3,
>
> We appreciate your positive comments on our manuscript (and a nice summary).
>
> __Non-image datasets.__
> As the reviewer pointed out, many of the algorithms that work well on the models for image datasets do not work particularly well on non-image models, due to the different training practices and/or model architectures. For this reason, we have reported additional experimental results on the language modeling task on Transformers in Appendix D of our original manuscript; we confirmed that LAMP continues to perform competitively, although the gain is smaller than in the image classification case. Regrettably, this point has not been emphasized enough in the original manuscript.
>
> In the revised draft, we make the following changes:
> - More pointers to Appendix D is given in the main text.
> - We slightly down-toned the abstract to remove exaggeration.
> - Enhanced Appendix D to have more discussions on the relevant work.
>
> ---
> __Typos.__
> Thank you for catching a few of these. They are now fixed!
>
> ---
>
> Please let us know if there are any more comments; we would be more than happy to continue the discussion!
>
> Sincerely,
> Authors.

---

> > ### Comment · AnonReviewer3 · 2020-11-24
> > **I like the changes**
> >
> > I like the change to the abstract which narrows the claim of superiority to apply only to image datasets. Much appreciated !
> >
> > I must confess I had missed Appendix D- that's great that it is included. It gives a sense for how the method performs on a non-image problem.

---

> > > ### Author Response · Authors · 2020-11-24
> > > **Thanks!**
> > >
> > > Dear reviewer,
> > >
> > > We are glad to hear that the revision we made was satisfactory to you. We really appreciate your additional effort to check our response and the revised manuscript.
> > >
> > > Best regards,
> > > Authors.

---

### Official Review · AnonReviewer1 · 2020-11-02
**Great Results; Section 3 Needs to be Rewritten**

**Rating:** 6
**Confidence:** 4

**Review:**

# After Rebuttal: Score Lowered from 7 to 6

## Concerns Addressed

I appreciate the effort the reviewers put into revising the paper to include the settings I suggested.

I am generally pleased with the revisions the authors made to the paper (especially Section 3), and I appreciate their attention to these details.

## Remaining Concern: Settings in the Main Body are Poorly-Tuned and May Overstate Results

I am concerned by one aspect of Figure 2: the unpruned accuracies for VGG-16 and ResNet-20 are much lower than they should be. VGG-16 should get 94% accuracy on CIFAR-10 (vs. 91% in the plot), and ResNet-20 should get 92% accuracy (vs. 86% accuracy in the plot).  This is because the paper uses Adam to train all networks without any learning rate drops, whereas the typical learning rate schedules for VGG-16 and ResNet-20 use SGD with momentum with learning rate drops.

This important difference raises the concern of whether the results shown in the paper will translate into fully-tuned, large-scale settings. As evidence of this concern, Appendix G does show a fully-tuned VGG-16 getting standard accuracy. In this setting, LAMP is no better than global magnitude pruning until very extreme sparsities.

**I have lowered my score on the basis that the results in Figure 2 may overstate the value of LAMP in well-tuned settings. I no longer have unequivocal confidence that LAMP is an improvement that should be adopted in general. I implore the authors to replace the experiments in Figure 2 with well-tuned versions of these networks that achieve SOTA accuracies.**

## Overall: Score Lowered from 7 to 6

I am less confident in the method's significance in well-tuned settings, and I can no longer unequivocally trust the empirical evaluation in the paper. I still support acceptance, but only tentatively.

# Overall

I think the technical results are excellent. The authors should be commended for making a productive contribution to the pruning literature. My biggest concerns are :
(1) The mathematical derivations are unreadable and impossible for future researchers to build on, so the authors *must* expand this section and make it crystal clear if I am to continue to recommend acceptance (they should use the extra page for this)
(2) The authors should include a network that isn't severely overparameterized for CIFAR-10. I recommend ResNet-20 or ResNet-56.
(3) The authors need to verify that their statements about prior papers are correct. I have noted a few things that need to be addressed below. In many cases, prior work was unclear about how it handled global/layerwise pruning decisions, and the authors should mention when there is ambiguity.

In addition, other smaller (but important) changes I would like to see include:
(4) The authors should ideally include results on all of ImageNet, since subsets of ImageNet tend to show very different results from the full task. However, I understand that this is very expensive and that not everyone has the resources to do so, and I understand if the authors are not in a position to accomplish this.
(5) The authors should use weight-rewinding as described below in the manner of "Linear Mode Connectivity and the Lottery Ticket Hypothesis" (Frankle et al.). Conv-6 is not a setting whose results correspond to larger-scale settings.
(6) I think the authors should revise the title to clarify that they are proposing a better way to select the layerwise rates for magnitude pruning. This paper makes a valuable contribution, and that should come through from the title so readers can determine this easily.

# Score

I have issued the paper a 7 on the assumption that the authors will make substantial revisions to Section 3 and take advantage of the extra page of space to do so. If they do not address my major concerns (and ideally my minor concerns), I will lower my score.

# Questions

Why is "minimizing the l2 distortion for the worst-case input signal" a reasonable design choice to make? The authors assert that it is what they aim to do, but they never explain why this is a good idea. I see why this makes sense at a high level, but adding a couple of sentences to this effect would be valuable, and I would like to hear from the authors directly.

"Minimizing l2 distortion for the worst-case input signal is equivalent to minimizing the spectral norm distortion." Please justify this. It's not immediate. I assume that the norm on the right side of (4) is the spectral norm, but please clarify.

"The optimization (4) can be relaxed to a Frobenius distortion minimization." Please justify this. It is not immediate. In general, this math is dense and the paper goes through it too quickly, and I wasn't able to follow it.

Where does the sparsity constraint k come from?

If I understand "MP: layerwise distortion minimization" correctly (which I don't think I do), it's meant to say that pruning the lowest-magnitude weights is a way of minimizing the l2 distortion for the worst-case input signal for a single layer in isolation? If this is the statement, I'm concerned that there's a <= in the derivation, since this suggests these two are not necessarily equivalent. And I don't understand why k is in there if we're pruning to a specific target sparsity. In general, I'm quite confused by (1) what you're trying to show here and (2) the actual details of this derivation.

The section on the LAMP score is completely uninformative. You need to include more of the derivation in the main body of the paper, especially explaining why it's appropriate to use the relaxation from the generalization theory literature (that again includes a <= suggesting that the two sides are not equivalent), what these quantities mean, what a "damage score" is, and how this leads to the LAMP scores. Right now, this paper suffers from - at best - obfuscation via math; worse, I highly doubt it would be possible for a reader (or someone working on follow-up work) to understand or reproduce your derivations/rationale. This is a crucial flaw in the paper.

The models you use are quite overparameterized in general. I would prefer to see a model like ResNet-20 or ResNet-56 on CIFAR-10, which is much less overparameterized for the specific task.

In general, the results look very impressive.

I am concerned that these graphs zoom in on the lowest sparsities. Importantly, it is difficult to tell what the unpruned accuracy looks like for these models.

For weight-rewinding, do not use Conv-6. It is a toy network with little correspondence to real behavior. Use a larger-scale network like VGG-1`6 or ResNet-20 and rewind to iteration 1000 as Frankle et al. do in "Linear Mode Connectivity and the Lottery Ticket Hypothesis."

# Things to Address

Verify that the details of the way that prior work has pruned networks is correct.

Ensure the purpose and derivations in Section 3.1 are explained in a way that is crystal clear to a reader. I'm a mathematically well-informed reader who is an expert on pruning, and I can't follow the derivations or what you're trying to say with those derivations. This is due to unclear writing; I'm not a poorly-informed reader.

# Other Notes

"The iterative pruning scheme of Han et al. uses global magnitude pruning." I believe this is incorrect. Han et al say that "the pruning threshold is chosen as a quality parameter multiplied by the standard deviation of a layer's weights."

"Frankle & Carbin (2019)...employ uniform layerwise sparsity." I believe this is incorrect. Frankle & Carbin use uniform sparsity for MNIST but use global pruning for larger-scale networks (ResNet-18 and VGG-19).

---

> ### Author Response · Authors · 2020-11-22
> **Response to R1: Part 1. Major concerns.**
>
> Dear R1,
>
> Thank you for taking your time and effort to take a careful look at our manuscript.
>
> ---
> __Clarity of Section 3__
> Thank you for this feedback. We revised Section 3 to make the following points clearer.
> (1) _Why distortion minimization:_ We speculate that the unexpected effectiveness of the MP comes from the fact that the MP solves a relaxed version of the layerwise output distortion minimization. Then, we use this speculation as a guiding principle to design a layerwise sparsity for MP (which might work unexpectedly well as if our speculation is true). The spirit is somewhat similar to the Optimal Brain Damage.
> (2) _Inequality in derivation:_ We now clearly state that MP solves a “relaxed” version of output distortion minimization.
> (3) _Sparsity constraint k:_ In the revised version, we state that the sparsity constraint is assumed to be coming from an operational constraint (e.g., required model size).
> (4) _Why is the used relaxation appropriate:_ The relaxation we used may not be the only relaxed form one can find. However, the relaxation provides several advantages, e.g. efficient computability.
> Please let us know if you believe that the revised section 3 can be clarified further. We would be more than happy to make any further revisions if necessary.
>
> ---
> __Smaller models for CIFAR-10.__
> Following the reviewer’s suggestion, we added an experimental result on ResNet-20 trained on CIFAR-10 (see **Figure 2-b**); the result replaces the previous result on ResNet-18. Similar to the ResNet-18 result, we observe that LAMP achieves the best performance. In addition, we observe that the accuracy of the ERK baseline closely matches the LAMP accuracy in this smaller model.
> ---
> __Descriptions of prior works__
> We thank the reviewer for pointing this out. We have corrected our descriptions on the algorithms of Han et al. (2015) and Frankle & Carbin (2019).

---

> ### Author Response · Authors · 2020-11-22
> **Response to R1: Part 2. Other concerns.**
>
> __Weight rewinding experiments__
> Following the reviewer’s suggestion, we conducted the weight rewinding experiment on VGG-16 following the settings stated in the “Linear Mode Connectivity and the Lottery Ticket Hypothesis” paper. The result is summarized in **Figure 4-b** of the revised manuscript. Similar to the Conv-6 case, we observe that LAMP performs the best. Unlike the Conv-6 experiment, however, ERK underperforms the global MP only in the low-to-medium sparsity regime, and outperforms global MP with higher sparsities. We have updated the discussions accordingly.
>
> ---
> __ImageNet experiments__
> To best respond to the reviewer’s question, we conducted _one-shot pruning_ on ResNet-18 trained on the ImageNet dataset, to the sparsity level of 10.74%. We decided to go with the one-shot pruning, as each pruning-retraining iteration took $~1.8$ days on our experimental settings; it was physically impossible for us to report any meaningful _iterative pruning_ result until the end of the response phase. Nevertheless, we will try to have the iterative pruning result ready for the final version.
>
> The experimental result is as follows: (figures denote the error rate.)
> unpruned [test@1 30.46] [test@5 10.84]
>
> lamp [test@1 41.40] [test@5 18.09]
> erk   [test@1 41.64] [test@5 18.41]
> glob [test@1 42.34] [test@5 18.98]
> unif [test@1 45.58] [test@5 21.46]
>
> ---
> __Zooming on lowest sparsities__
> The “zoom ratios” of the figures in the original manuscript were selected to capture both the sparsity levels without any performance loss and the sparsity levels where the performance differences are more visible. On the other hand, we agree that there is a room for further clarification.
>
> To address your concern, we have added **two dotted horizontal lines** to denote the (1) peak accuracy among all sparsity levels and (2) the accuracy at zero sparsity. If you have any more suggestions on how to visualize even better, we would be more than happy to incorporate.
>
> ---
> Please let us know if you have any further concerns.
>
>
> Sincerely,
> Authors.

---

### Author Response · Authors · 2020-11-12
**An early response**

Dear reviewers and AC,

We wholeheartedly appreciate your insightful comments and constructive suggestions provided to help us improve our manuscript. Embracing the feedback, we are making several revisions and conducting supplementary experiments, which we hope to deliver soon.

Before the upcoming revision, however, we would like to briefly address some of the concerns which might have come from a slight misunderstanding; we would provide detailed clarifications via individual replies to each reviews.

Best regards,
Authors.

---

### Author Response · Authors · 2020-11-22
**Summary of revisions**

Dear Reviewers and AC,

Thank you again for your continuing time and effort to provide detailed feedback on our manuscript. To best appreciate your comments, we conducted several supplementary experiments and revised the manuscript; here is a short list of updates.

- Added experiment on ResNet-20 (less overparameterized model) (R1).
- Added weight-rewinding experiment on VGG-16 (R1).
- Fixed the misleading descriptions on the prior work (R1, R2).
- Revised Section 3 with more detailed explanations (R1, R2).
- Updated figures with additional visual cues for unpruned/peak accuracies (R1)
- Added experiments with state-of-the-art training recipe of VGG-16 (R2)
- Added comparisons to the layerwise sparsity discovered by AMC (R2)
- … and other editorial revisions and added explanations to enhance the clarity of the manuscript (R1,2,3,4).

The updated texts--except for editorial changes--are marked in **purple.**

Via individual comments, we give detailed responses and clarifications to the concerns raised by the reviewers, with pointers to corresponding revisions of the manuscript.

Best regards,
Authors.

---

### Author Response · Authors · 2020-11-24
**Title change.**

Dear reviewers and AC,

Following R1's suggestion, we just changed the title of our manuscript from "A deeper look at the layerwise sparsity of magnitude-based pruning" to "Layer-adaptive sparsity for the magnitude-based pruning." We believe that this change would make it easier for the readers to grasp the main contribution of our paper. Any other comments on the title would be greatly appreciated.

Best,
Authors.

---

### Decision · Program_Chairs · 2021-01-07
**Final Decision**

**Decision:**

Accept (Poster)

**Comment:**

The paper proposes a layer-wise magnitude-based tuning method through the adoption of LAMP score, motivated by minimizing the model output distortion. The new importance score differs from vanilla magnitude-based score in that it incorporate more layer-wise information. Extensive experiments are conducted on image and language models to show the improved accuracy upon prior arts under same model compression ratio. Ablation study is also provided to further explain the intuition and comparison of LAMP with other pruning methods.

Though the experiments are extensive, some reviewers raised questions that only image datasets are tested. In the rebuttal, the authors addressed more on Appendix D which provides non-image results, and also modified the abstract to highlight the efficacy on image data. In all, given the extensive empirical evaluation on various datasets and model architectures, the improvement of LAMB over prior methods seems convincing. Nevertheless, we urge the authors to include more experimental results, for example ResNet-18 on ImageNet as promised to Reviewer 1, to make the results more solid. It is also suggested to include and discuss some relevant papers mentioned by the reviewers in the final version.